# *Fusobacterium nucleatum* reduces METTL3-mediated m⁶A modification and contributes to colorectal cancer metastasis

Shujie Chen, [1,2,3,4,9], Lu Zhang[1,3,4,9], Mengjie Li[1,3,4,9], Ying Zhang[1,2,3,4,9], Meng Sun[1,3,4], Lingfang Wang[5], Jiebo Lin[5], Yun Cui[1,3,4], Qian Chen[1,3,4], Chenqi Jin[1,3,4], Xiang Li[1,3,4], Boya Wang[3,6], Hao Chen[7], Tianhua Zhou [1,3,4✉], Liangjing Wang [3,4,8✉], Chih-Hung Hsu [5✉] & Wei Zhuo [1,3,4✉]

Microbiota-host interactions play critical roles in colorectal cancer (CRC) progression, however, the underlying mechanisms remain elusive. Here, we uncover that *Fusobacterium nucleatum* (*F. nucleatum*) induces a dramatic decline of m⁶A modifications in CRC cells and patient-derived xenograft (PDX) tissues by downregulation of an m⁶A methyltransferase METTL3, contributing to induction of CRC aggressiveness. Mechanistically, we characterized forkhead box D3 (FOXD3) as a transcription factor for METTL3. *F. nucleatum* activates YAP signaling, inhibits FOXD3 expression, and subsequently reduces METTL3 transcription. Downregulation of METTL3 promotes its target kinesin family member 26B (KIF26B) expression by reducing its m⁶A levels and diminishing YTHDF2-dependent mRNA degradation, which contributes to *F. nucleatum*-induced CRC metastasis. Moreover, METTL3 expression is negatively correlated with *F. nucleatum* and KIF26B levels in CRC tissues. A high expression of KIF26B is also significantly correlated with a shorter survival time of CRC patients. Together, our findings provide insights into modulating human m⁶A epitranscriptome by gut microbiota, and its significance in CRC progression.

[1] Department of Cell Biology and Department of Gastroenterology of Sir Run Run Shaw Hospital, Zhejiang University School of Medicine, Hangzhou, China. [2] Department of Gastroenterology, Sir Run Run Shaw Hospital, Zhejiang University School of Medicine, Hangzhou, China. [3] Institute of Gastroenterology, Zhejiang University, Hangzhou, China. [4] Cancer Center, Zhejiang University, Hangzhou, China. [5] Women's Hospital, Institute of Genetics, and Department of Environmental Medicine, Zhejiang University School of Medicine, Zhejiang, China. [6] Department of Pharmacy, Sir Run Run Shaw Hospital, Zhejiang University School of Medicine, Hangzhou, China. [7] School of Medicine, Southern University of Science and Technology, Shenzhen, China. [8] Department of Gastroenterology, Second Affiliated Hospital of Zhejiang University School of Medicine, Hangzhou, China. [9] These authors contributed equally: Shujie Chen, Lu Zhang, Mengjie Li, Ying Zhang. ✉email: tzhou@zju.edu.cn; wangljzju@zju.edu.cn; ch_hsu@zju.edu.cn; wzhuo@zju.edu.cn

Colorectal cancer (CRC) is the fourth most commonly diagnosed cancer worldwide and the second leading cause of cancer-related death[1]. Despite advances in the diagnosis and treatment of CRC, the 5-year survival rate remains low because of tumor metastasis or recurrence[2]. Therefore, it is of paramount importance to elucidate the underlying mechanisms of metastasis in CRC patients.

Recent studies have indicated that microbiota–host interactions play important roles in the progression of CRC. With the development of metagenomics sequencing, a variety of microbiota involved in the carcinogenesis of CRC have been revealed, such as *Fusobacterium nucleatum (F. nucleatum)*[3], *Peptostreptococcus anaerobius*[4], and *Akkermansia muciniphila*[5]. *F. nucleatum* is a gram-negative aerobic bacterium prevalent in the oral cavity whose abundance has been found to be increased in colorectal carcinoma[6,7]. *F. nucleatum* lectin Fap2 is able to recognize the host Gal-GalNAc and help this bacterium abundantly localize in the tissues of CRC and colonic adenomas[8]. An abnormal proportion of *F. nucleatum* generates a pro-inflammatory microenvironment[9], increases the proliferation of CRC cells[10], and promotes chemoresistance of CRC[11]. However, the underlying mechanisms of *F. nucleatum* in CRC metastasis remain unclear.

There is growing evidence that bacterial pathogens modulate host epigenomics, including DNA methylation, histone methylation, and acetylation[12–15]. Very recently, the presence of the microbiome has been suggested to induce changes of epitranscriptomic modifications in the tissues of mice[16,17]. Epitranscriptomic modification, most notably N6-methyladenosine (m6A), has emerged as an important regulator of mRNA to influence various fundamental bioprocesses[18]. The dysregulation of m6A modification and its associated proteins has shown significant effects on human carcinogenesis. METTL3, the major m6A methyltransferase, is involved in many cancers, including acute myeloid leukemia[19], hepatocellular carcinoma[20], and lung cancer[21]. A number of downstream targets of METTL3 have been reported; however, the upstream mechanisms regulating METTL3 are still unclear. Emerging studies show that m6A modification of mRNA is a highly dynamic process influenced by stress responses[22], the circadian rhythm[23], viral infection[24,25], and microbiota infections[16,17]. Whether microbiota–host interactions influence the m6A modification of host mRNA and its underlying mechanisms remain elusive.

In this study, we report that *F. nucleatum* induces a significant decrease in m6A levels in human CRC cells and in patient-derived xenograft (PDX) tissues. Mechanistically, *F. nucleatum* activates yes1 associated transcriptional regulator (YAP) signaling to inhibit forkhead box D3 (FOXD3), which we identified as a positive transcription factor of methyltransferase-like 3 (METTL3). *F. nucleatum* treatment downregulates METTL3 expression and inhibits YTHDF2-mediated degradation of kinesin family member 26B (KIF26B) mRNA. We demonstrated that KIF26B is essential for *F. nucleatum*-induced CRC aggressiveness and metastasis. Collectively, this work reveals that *F. nucleatum* reduces m6A modifications to promote CRC aggressiveness through the YAP/FOXD3/METTL3/KIF26B axis.

## Results

### *F. nucleatum* reduces METTL3-mediated m6A modifications in human CRC cells and PDX tissues

To study the effects of *F. nucleatum* on the m6A epitranscriptomic modifications of human CRC cells, we treated HCT116 cells with *F. nucleatum*, *Escherichia coli* (*E. coli*) DH5α, or PBS control for 24 h. Dot blot analysis revealed that the m6A levels in cells cocultured with *F. nucleatum* were robustly decreased compared to those in the

*E. coli*-treated cells or control cells (Fig. 1a). The downregulation of m6A modifications by *F. nucleatum* was also observed in another CRC cell LoVo (Supplementary Fig. 1a). Interestingly, short treatment with *F. nucleatum* was able to induce the decrease in mRNA m6A modifications (Fig. 1b). More importantly, the significant decrease of the m6A in total mRNA in *F. nucleatum*-treated CRC cells was also confirmed by ultra-high-performance liquid chromatography coupled with quadrupole-Exactive mass spectrometry (UHPLC Q-Exactive MS) analysis (Fig. 1c). Consistent with our previous findings[26], *F. nucleatum* treatment dramatically promoted the migration (Fig. 1d and Supplementary Fig. 1c) and invasion (Supplementary Fig. 1b and Supplementary Fig. 1d) of both HCT116 cells and LoVo cells compared with *E.coli* or PBS treatment.

It has been well documented that m6A writers and erasers participate in the dynamic homeostasis of m6A modification. Western blot analysis showed that *F. nucleatum* treatment obviously reduced the expression of METTL3 in both HCT116 cells (Fig. 1e and Supplementary Fig. 1e) and LoVo cells (Supplementary Fig. 1f, g), while *F. nucleatum* treatment did not visibly influence the expression of other m6A erasers or readers (Fig. 1e and Supplementary Fig. 1f), suggesting that the decrease in m6A modifications may result from the downregulation of METTL3 by *F. nucleatum*.

Furthermore, we established a PDX model generated from CRC patients (Fig. 1f). The PDX tumor tissues treated with *F. nucleatum* ex vivo showed an obvious decrease in global m6A levels (Fig. 1g). The *METTL3* mRNA levels were also significantly reduced in *F. nucleatum*-treated PDX tissues compared to the tissues treated with *E. coli* or PBS control (Fig. 1h).

METTL3 has been reported to promote translation of certain mRNAs independently of its m6A methyltransferase activity[21]. To examine whether the m6A methyltransferase activity of METTL3 contributes to the reduced m6A levels induced by *F. nucleatum*, plasmids expressing wild-type METTL3, or its catalytic mutant (aa395–398 and DPPW-APPA) were constructed and applied to the *F. nucleatum*-treated cells (Supplementary Fig. 1h). Ectopic expression of wild-type METTL3, but not its catalytic mutant, rescued the decreased m6A levels caused by *F. nucleatum* (Fig. 1i). Moreover, we tried to determine if METTL3-mediated m6A modification is involved in the enhanced migration of CRC cells induced by *F. nucleatum* treatment. Consistent with the dot blot results, the transwell assays showed that overexpression of wild-type METTL3, but not its catalytic mutant, obviously reversed the enhanced migration of HCT116 cells stimulated by *F. nucleatum* treatment (Fig. 1j). Taken together, our data indicate that *F. nucleatum* decreases the global m6A modification levels of human CRC cells and PDX tissues to promote CRC cell aggressiveness by downregulating METTL3.

### *F. nucleatum* inhibits METTL3 by modulating the Hippo-YAP signaling pathway

To explore the mechanism of how *F. nucleatum* inhibits METTL3 and further modulates the homeostasis of m6A modification, we screened for signaling pathways activated in CRC cells upon *F. nucleatum* treatment. The target genes related to various signaling pathways that have been reported to play important roles in CRC progression and metastasis were detected by quantitative real-time PCR (qRT-PCR). We found that the genes related to NF-κB and the Hippo-YAP signaling pathways were activated in both HCT116 and LoVo cells (Fig. 2a, b). We isolated the cytoplasmic and nuclear components of HCT116 cells and found that *F. nucleatum* increased the proportion of YAP in the nucleus (Fig. 2c). Immunofluorescence assay results also corroborated the significantly increased YAP

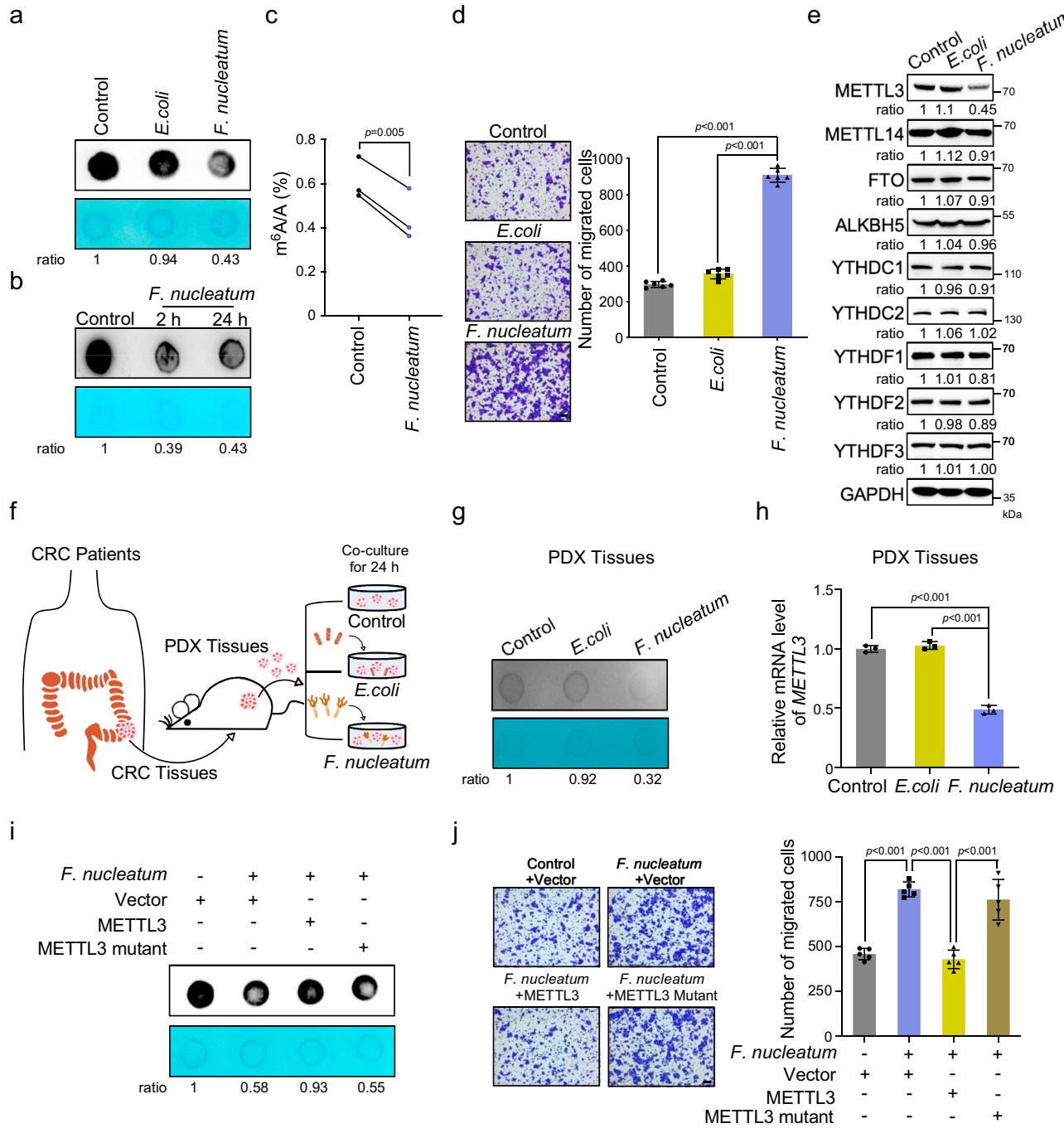

**Fig. 1 *F. nucleatum* reduces METTL3-mediated m⁶A modifications in human CRC cells and PDX tissues. a** mRNA dot blot analysis was performed to determine the m6A levels of HCT116 cells treated with *F. nucleatum*, *E.coli* DH5α, or PBS control for 24 h. **b** mRNA dot blot analysis was performed to determine the m6A levels of HCT116 cells treated with *F. nucleatum* for 2 or 24 h. **c** UHPLC Q-Exactive MS analysis was performed to determine the m6A/A ratio of the total mRNA in HCT116 cells treated with *F. nucleatum* or PBS control for 24 h. **d** HCT116 cells were pretreated with *F. nucleatum*, *E.coli* DH5α, or PBS control for 2 h and subjected to transwell assay (Left). The migrated cell were quantified by counting in six fields (Right). Scale bar, 100 μm. **e** Western blot was performed to determine the m6A modification-associated protein levels in HCT116 cells with indicated treatment. **f** Schematic illustration of the PDX model established with tumor tissues from CRC patients. The ex vivo model of PDX tissues treated with *F. nucleatum*, *E.coli* DH5α, or PBS control was shown. **g, h** The PDX tissues treated with *F. nucleatum*, *E.coli* DH5α, or PBS control for 24 h were subjected to mRNA dot blot analysis of m6A levels (**g**). Quantitative RT-PCR analysis was performed to detect the expression of *METTL3* in PDX tissues (**h**). **i** The HCT116 cells were transfected with wild-type METTL3, or catalytic mutant (aa395–398, DPPW-APPA) METTL3, or control pcDNA3.1(+) plasmids and treated with *F. nucleatum* or PBS. mRNA dot blot analysis of m6A levels were performed. **j** The HCT116 cells with indicated treatments were applied for transwell migration analysis. Representative images of migrated cells were shown (Left). The migrated cells were quantified in five fields (Right). Scale bar, 100 μm. The methylene blue staining was used as a loading control in the mRNA dot blot assay. Data were from one representative of three independent experiments (**a**, **b**, **e**, **g**, **i**). Data were shown as mean ± SD. *P* values were shown. A two-tailed paired *t*-test (**c**), two-tailed Student's *t*-test (**d**, **h**, **j**).

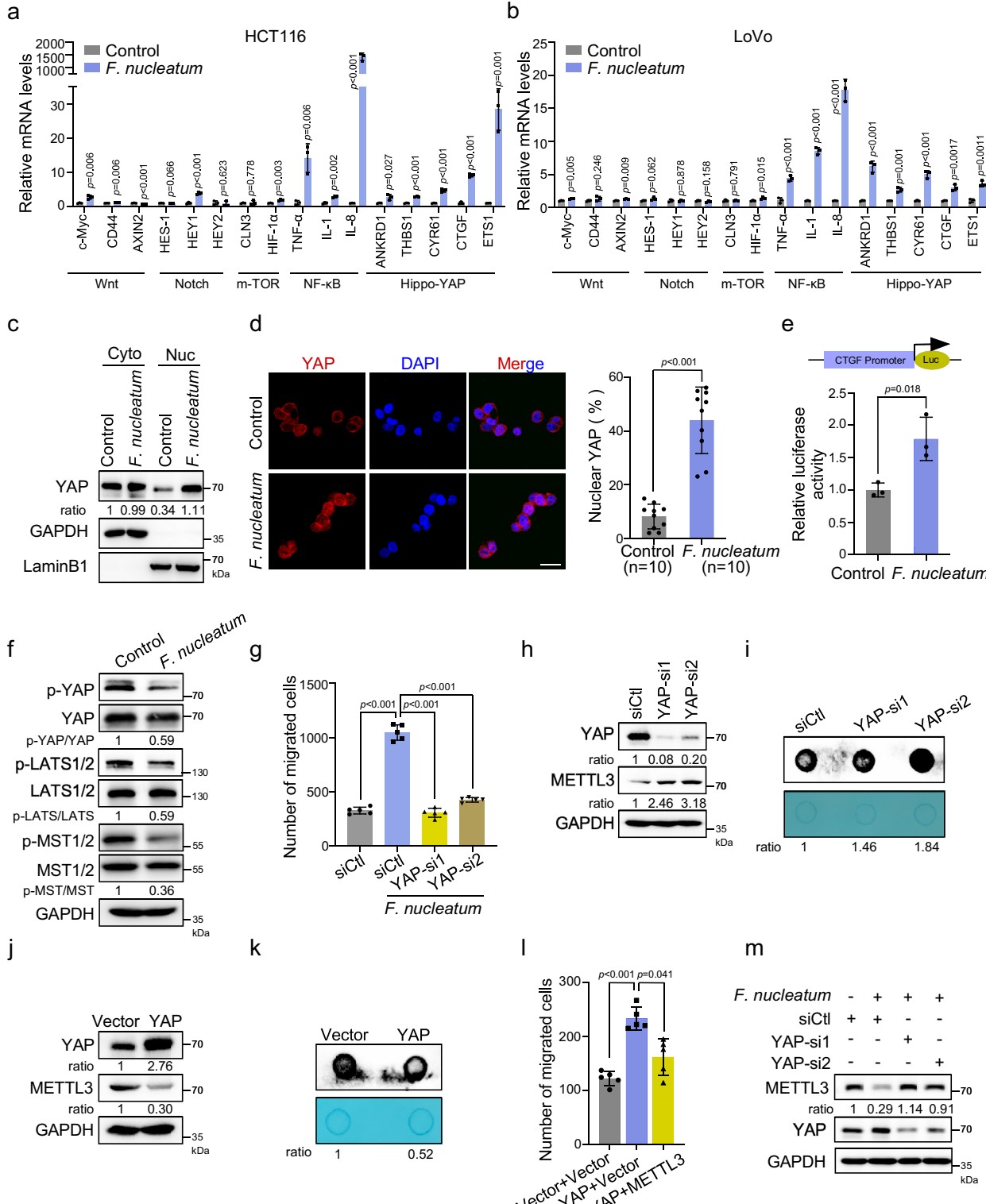

levels in the nucleus of *F. nucleatum*-treated cells (Fig. 2d). Dual-luciferase reporter assay[27] further confirmed the activation of YAP signaling in CRC cells by *F. nucleatum* (Fig. 2e).

Given that the Hippo pathway is a prime input for YAP activity[28], we performed western blot analysis to examine the phosphorylation of MST1/2 and LATS1/2 and found their phosphorylation levels were obviously reduced in the *F. nucleatum*-treated HCT116 and LoVo cells (Fig. 2f and Supplementary Fig. 2a). Moreover, we tried to analyze the

upstream events of the Hippo pathway. NF2, KIBRA, and WILLIN\FRMD6 are well-known classical activators for MST1/2[29,30]. Interestingly, *F. nucleatum* treatment notably inhibited the levels of these three activators in both HCT116 and LoVo cells (Supplementary Fig. 2b, c). Collectively, these results suggest that *F. nucleatum* treatment actives YAP signaling may through inhibition of the Hippo pathway in CRC cells.

We next investigated whether the YAP signaling mediates the function of *F. nucleatum* in promoting CRC aggressiveness.

**Fig. 2 _F. nucleatum_ inhibits METTL3 through modulating Hippo-YAP signaling pathway. a, b** Quantitative RT-PCR was performed to detect the variations of different pathway target genes in HCT116 cells (**a**) and LoVo cells (**b**) treated with _F. nucleatum_ or PBS control for 2 h. **c** Western blot analysis of the levels of YAP in the cytoplasmic and nuclear fractions of HCT116 cells treated with _F. nucleatum_ or PBS control for 2 h. GAPDH, cytoplasm marker. LaminB1, nuclear marker. **d** Immunofluorescence analysis of the YAP distribution in the indicated cells (Left). Cells were stained with a specific antibody against YAP (red), and nuclei were counterstained with DAPI (blue). The percentage of nuclear YAP were quantified by counting in ten fields (Right). Scale bar, 20 μm. **e** Dual-luciferase reporter assay showing the effects of _F. nucleatum_ treatment on relative _CTGF_-promoter activity in the HCT116 cells. **f** Western blot was performed to detect the levels of YAP and phospho-YAP, LATS1/2 and phospho-LATS1/2, MST1/2 and phospho-MST1/2 in HCT116 cells treated with _F. nucleatum_ or PBS control. **g** Transwell migration assay was performed in HCT116 cells with indicated treatment. The migrated cells were quantified by counting in five fields. **h** HCT116 cells with the indicated treatment were subjected to western blot analysis. **i** mRNA dot blot analysis was performed to determine the m6A levels of HCT116 cells after transfection with siRNAs targeting YAP. **j** HCT116 cells transfected with YAP overexpression or control pcDNA3.1(+) plasmids were subjected to western blot analysis of METTL3. **k** mRNA dot blot analysis was performed to determine the m6A levels of HCT116 cells after transfection with YAP plasmid or pcDNA3.1(+) control vector. **l** Transwell migration analysis of HCT116 cells transfected the indicated vectors. The migrated cells were quantified by counting in five fields. **m** Western blot analysis of METTL3 in HCT116 cells transfected with the indicated siRNAs. The methylene blue staining was used as a loading control in the mRNA dot blot assay. Data were from one representative of three independent experiments (**c**, **f**, **h**–**k**, **m**). Data were shown as mean ± SD. _P_ values were shown. Two-tailed Student's _t_-test (**a**, **b**, **d**, **e**, **g**, **l**).

Transwell assay revealed that knockdown of YAP significantly reversed the enhanced cell migration ability of HCT116 cells or LoVo cells induced by _F. nucleatum_ (Fig. 2g and Supplementary Fig. 2d). These results further prompted us to investigate whether YAP regulates METTL3 expression. METTL3 protein levels and m6A levels were significantly increased when YAP was silenced by two different siRNAs (Fig. 2h, i). Conversely, ectopic expression of YAP downregulated METTL3 levels and m6A levels, which resembled the effect of _F. nucleatum_ on CRC cells, suggesting that YAP signaling may be a negative upstream regulator of METTL3 in CRC cells (Fig. 2j, k). Indeed, overexpression of METTL3 inhibited the cell migration induced by ectopic expression of YAP (Fig. 2l). Moreover, we verified the role of YAP in regulating the expression of METTL3 in the presence of _F. nucleatum_. Our data showed that _F. nucleatum_-treatment decreased the expression of METTL3, while depletion of YAP almost entirely rescued the downregulation of METTL3 expression induced by _F. nucleatum_ (Fig. 2m), implying that YAP signaling is involved in _F. nucleatum_-induced downregulation of METTL3 expression in CRC cells.

Although the NF-κB pathway was activated by _F. nucleatum_, similar to a previous report[10], we observed that knockdown of NF-κB did not influence METTL3 expression (Supplementary Fig. 2e). Furthermore, we treated HCT116 cells with an inhibitor of NF-κB, BAY11-7082, and discovered that BAY11-7082 inhibited the activation of NF-κB signaling, but not reversed the _F. nucleatum_-induced downregulation of METTL3 (Supplementary Fig. 2f). Collectively, our data suggest that _F. nucleatum_ activates YAP signaling by blocking the Hippo cascade, leading to reduced METTL3 levels in CRC cells.

**FOXD3 is a transcription factor for METTL3**. We sought to determine the mechanism of YAP signaling controlling METTL3 expression. YAP is known to regulate gene transcription through its association with the TEAD transcription factors[28]. However, we failed to find binding sites of TEADs in the promoter of METTL3. Instead, using the online bioinformatics tools _AnimalTFDB_, _TRANSFAC_, and _JASPAR_, we predicted the potential transcription factors that may bind to the promoter regions of METTL3. The Venn diagram revealed that there were many binding sites of FOXD3 in the promoter region of METTL3 (−1778 to −1957 bp), implicating FOXD3 as a potential transcription factor for METTL3 (Fig. 3a).

The ectopic expression of FOXD3 significantly increased the mRNA (Supplementary Fig. 3a) and protein (Fig. 3b) levels of METTL3. Conversely, the mRNA (Supplementary Fig. 3b) and protein (Fig. 3c) levels of METTL3 were significantly reduced

when FOXD3 was knocked down. Furthermore, overexpression of FOXD3 increased m6A levels (Fig. 3d) while depletion of FOXD3 decreased the m6A levels (Fig. 3e), suggesting that FOXD3 is a positive regulator for METTL3. To verify whether FOXD3 is a direct transcription factor for METTL3, chromatin immunoprecipitation (ChIP) assays were employed, and the results revealed that FOXD3 bound to the METTL3 promoter; in the regions of FOXD3-Primer1 and FOXD3-Primer2, both of which may contain putative FOXD3 binding elements (Fig. 3a, f). Furthermore, a dual-luciferase reporter assay confirmed that the overexpression of FOXD3 significantly enhanced METTL3-luciferase activity in HCT116 cells (Fig. 3g). These results indicate that FOXD3 is a positive transcription factor for METTL3.

Interestingly, ChIP assays showed that _F. nucleatum_ treatment significantly reduced the interaction of FOXD3 with the promoter region of METTL3, which suggests _F. nucleatum_ treatment repressed FOXD3-mediated transcription of METTL3 (Fig. 3h). Dual-luciferase reporter assays confirmed that _F. nucleatum_ treatment indeed tempered FOXD3-stimulated transcriptional activity of METTL3 (Fig. 3i). We further found that _F. nucleatum_ treatment significantly downregulated FOXD3 levels in PDX tissues (Fig. 3j). A similar result was also observed in the _F. nucleatum_-treated HCT116 and LoVo cells (Fig. 3k and Supplementary Fig. 3c). Next, we knocked down the expression of YAP in HCT116 cells and found FOXD3 was increased (Fig. 3l). Western blot assays confirmed that the depletion of YAP increased the expression of FOXD3 (Fig. 3m). Collectively, our work reveals FOXD3 as a transcription factor for METTL3. _F. nucleatum_ downregulates METTL3 may through YAP/FOXD3 axis.

**Variations of m6A-regulated genes in CRC cells with _F. nucleatum_ treatment**. To investigate the variations of m6A modifications in specific genes, we performed transcriptome-wide m6A sequencing in _F. nucleatum_-treated HCT116 cells. m6A sequencing analysis identified 5841 m6A peaks that were changed in _F. nucleatum_-treated HCT116 cells compared to the untreated cells, including 2835 m6A-Up peaks and 3006 m6A-Down peaks. A special WGGAM (W = A/T, M = A/C) motif was highly enriched within m6A sites in the untreated group, while a VWGGA (V = C/G/A, W = A/T) motif was present in _F. nucleatum_-treated cells (Fig. 4a). Moreover, we observed that m6A peaks were especially abundant in the vicinity of start and stop codons. An obvious decrease in m6A peaks was found in _F. nucleatum_-treated cells (Fig. 4b). Based on the m6A-seq results, the total m6A distribution patterns of mRNAs were further analyzed. We observed that the m6A peaks in the vicinity of

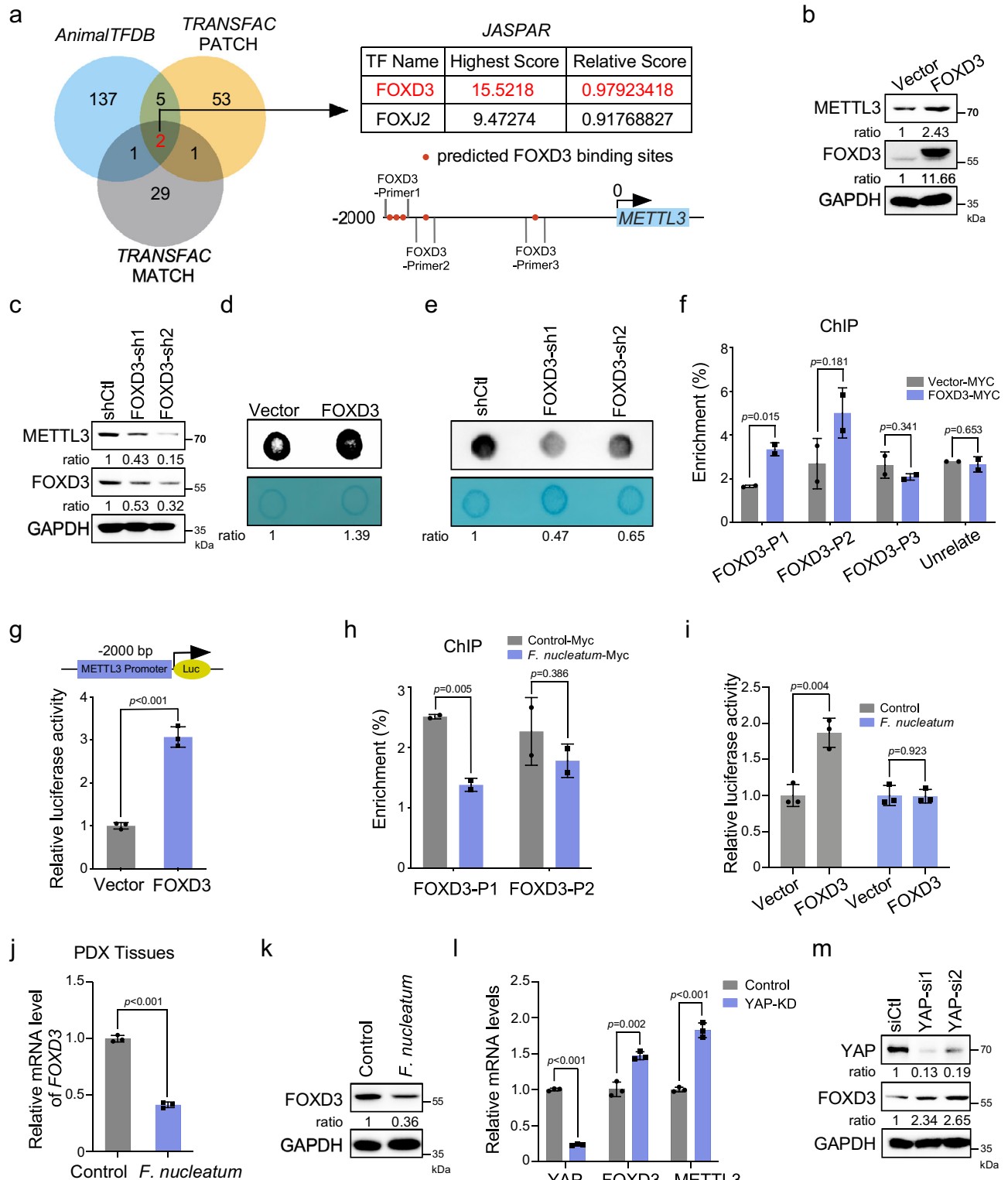

5′UTR, and CDS regions of mRNAs in *F. nucleatum*-treated cells were decreased, although the 3′UTR region maintained a similar abundance (Fig. 4c).

A total number of 3589 genes were identified in *F. nucleatum*-treated cells, which showed a significant differential change in $m^6A$ levels. Among these genes, the $m^6A$ levels of 1585 genes were upregulated and 2004 genes were downregulated. Based on these differential genes, KEGG pathway enrichment analysis revealed that *F. nucleatum* impacted multiple aspects of CRC cells, including focal adhesion, tight junction, and Hippo signaling pathways (Fig. 4d and Supplementary Fig. 4a, b), which was consistent with our previous data showing a correlation between *F. nucleatum* and Hippo-YAP signaling. In addition, gene ontology (GO) enrichment analysis revealed that $m^6A$ modifications occurred on a handful of genes related to the regulation of GTPase activity, cytoskeleton organization, and cell polarity remodeling in *F. nucleatum*-treated CRC cells (Fig. 4e and Supplementary Fig. 4c, d).

**Fig. 3 FOXD3 is a transcription factor for METTL3. a** Venn diagram showing two overlapping transcription factors appeared in three prediction sets (*AnimalTFDB*, *TRANSFAC* PATCH, and *TRANSFAC* MATCH). Schematic illustration of the potential binding sites of FOXD3 on the promoter of METTL3. Three pairs of primers were shown for the following ChIP analysis. **b** Western blot analysis of METTL3 in HCT116 cells transfected with FOXD3-myc-his vector or control vector. **c** Western blot analysis of METTL3 in FOXD3-knockdown HCT116 cells. **d, e** mRNA dot blot analysis was performed to determine the $m^6A$ levels of HCT116 cells transfected with FOXD3 plasmid (**d**) or FOXD3 shRNAs (**e**). **f** ChIP-qPCR analysis of FOXD3 binding to the predicted binding regions of METTL3 promoter in HCT116 cells. Ectopic FOXD3 was pulled down by the anti-MYC-tag antibody. **g** Dual-luciferase reporter assay showing the effects of FOXD3 overexpression on relative *METTL3*-promoter ($-2000$ bp ~$100$ bp) activity in the HCT116 cells. **h** ChIP-qPCR analysis of FOXD3 binding to the promoter of METTL3 in HCT116 cells treated with *F. nucleatum* or PBS control. **i** Dual-luciferase reporter assay showing the effects of FOXD3 on relative *METTL3*-promoter activity in the HCT116 cells treated with *F. nucleatum* or PBS control. **j** CRC PDX tissues were treated with *F. nucleatum* or PBS for 24 h and subjected to quantitative RT-PCR analysis of *FOXD3* expression. **k** Western blot analysis of FOXD3 expression in HCT116 cells treated with *F. nucleatum* or PBS control. **l** Quantitative RT-PCR analysis of *FOXD3* and *METTL3* expression in HCT116 cells transfected with indicated siRNAs. **m** Protein levels of FOXD3 were detected in YAP-knockdown HCT116 cells. The methylene blue staining was used as a loading control in mRNA dot blot assay (**d**, **e**). The ChIP-qPCR results were presented as an enrichment of FOXD3 at METTL3 promoter relative to input (**f**, **h**). Data were from one representative of three independent experiments (**b**–**e**, **k**, **m**). Data were shown as mean ± SD. *P* values were shown. Two-tailed Student's *t*-test (**f**, **g**, **h**, **i**, **j**, **l**).

**KIF26B is a downstream target of METTL3 by $m^6A$-seq and RNA-seq**. To characterize potential downstream targets involved in $m^6A$-regulated CRC aggressiveness stimulated by *F. nucleatum*, we performed mRNA-sequencing in *F. nucleatum*-treated HCT116 with independent biological replicates. The expression profiling analysis identified 155 upregulated and 88 downregulated genes (Fig. 5a). We identified five genes overlapping between 2004 $m^6A$-downregulated genes and 155 mRNA-upregulated genes (Fig. 5b). Among the five identified genes, kinesin family member 26B (KIF26B) is involved in cytoskeleton reorganization, which is important for CRC aggressiveness, as presented by GO analysis (Fig. 4e). $m^6A$ sequencing showed a significant decrease in $m^6A$ peaks around the stop codon of *KIF26B* mRNA upon *F. nucleatum* treatment (Fig. 5c). More importantly, we designed primers for $m^6A$-RT-qPCR in the region containing the changed $m^6A$ peak, which were annotated in our schema diagram (Fig. 5c). The $m^6A$-RT-qPCR assay was performed and the result showed that the $m^6A$ enrichment of *KIF26B* mRNA was indeed reduced upon *F. nucleatum* treatment (Fig. 5d). In the meanwhile, our data confirmed that both the mRNA and protein levels of KIF26B were significantly increased in *F. nucleatum*-treated HCT116 (Fig. 5e, f) and LoVo cells (Supplementary Fig. 5a, b). Therefore, we selected KIF26B as a candidate target of METTL3-mediated $m^6A$ modification in CRC cells.

To validate that *KIF26B* mRNA is a target of METTL3, RNA immunoprecipitation (RIP) assays were applied, in which METTL3 target gene *CREBBP* was served as positive control while *HPRT1* was served as a negative control[31]. Our data showed that there was an obvious interaction between *KIF26B* mRNA and METTL3, which was comparable to the interaction between METTL3 and its well-known target *CREBBP* mRNA, suggesting *KIF26B* mRNA as a direct target of METTL3 (Fig. 5g). Knockdown of METTL3 significantly increased the expression of KIF26B in HCT116 (Fig. 5h, i) and LoVo cells (Supplementary Fig. 5c, d). Moreover, downregulation of YAP significantly decreased the protein levels of KIF26B (Supplementary Fig. 5e), while the knockdown of FOXD3 significantly induced KIF26B expressing (Supplementary Fig. 5f). These data suggested that the expression of KIF26B is regulated by the YAP/FOXD3/METTL3 axis. Importantly, *F. nucleatum*-induced upregulation of KIF26B was reversed by ectopic expression of wild-type METTL3, but not its catalytic mutant, suggesting that upregulation of KIF26B by *F. nucleatum* is dependent on the $m^6A$ methyltransferase activity of METTL3 (Fig. 5j).

To further confirm that $m^6A$ modifications contribute to the regulation of KIF26B expression, we treated HCT116 cells with the global methylation inhibitor 3-deazaadenosine (DAA)[20].

DAA treatment induced an obvious increase in KIF26B mRNA and protein expression (Fig. 5k, l). Collectively, our findings indicate that *F. nucleatum* treatment induces the upregulation of KIF26B through METTL3-mediated $m^6A$ modifications.

The substantially increased mRNA levels of *KIF26B* by *F. nucleatum* treatment or METTL3 knockdown prompted us to detect the mRNA stability of *KIF26B*. *F. nucleatum*-treated HCT116 and LoVo cells were treated with actinomycin D to block transcription. qRT-PCR showed that *F. nucleatum* significantly prolonged the half-life of *KIF26B* mRNA in both HCT116 and LoVo cells (Fig. 5m and Supplementary Fig. 5g). Previous studies have reported that YTHDF2 specifically recognizes $m^6A$-modified mRNAs to modulate their stability[31,32]. Accordingly, we employed RIP assays and observed that YTHDF2 is bound to *KIF26B* mRNA (Fig. 5n). Depletion of YTHDF2 induced a significant increase in *KIF26B* mRNA in YTHDF2-depleted HCT116 cells (Fig. 5o) while silencing of YTHDF1 or YTHDF3 had no significant effect on the expression of *KIF26B* mRNA (Supplementary Fig. 5h, i).

Taken together, these results suggest that *F. nucleatum*-induced downregulation of METTL3 reduces $m^6A$ modification levels of KIF26B and further diminishes YTHDF2-dependent mRNA degradation, thereby facilitating KIF26B expression in CRC cells.

***F. nucleatum* accelerates CRC aggressiveness and metastasis by upregulating KIF26B**. Recent studies have reported that KIF26B is involved in gastric cancer and breast cancer as an oncogene[33,34]. However, the potential roles of KIF26B in CRC aggressiveness and metastasis have not been studied. We knocked down KIF26B in HCT116 cells and performed mRNA-sequencing (Fig. 6a). Transcriptome profiling revealed that KIF26B was associated with cell-cell junctions and response to growth factor stimulation (Fig. 6b), which are important for CRC cell aggressiveness. Intriguingly, gene set enrichment analysis and KEGG pathway enrichment analysis showed that the Hippo signaling pathway was significantly enriched in KIF26B-regulated gene signatures (Fig. 6c and Supplementary Fig. 6a).

Next, we depleted KIF26B in HCT116 cells with two independent siRNAs (Supplementary Fig. 6b). Knockdown of KIF26B dramatically inhibited the migration and invasion of HCT116 cells (Supplementary Fig. 6c, d). Similar results were also observed in LoVo cells (Supplementary Fig. 6e–g). Moreover, KIF26B was knocked down in *F. nucleatum*-treated HCT116 cells or LoVo cells. As expected, *F. nucleatum* obviously promoted cell migration, while knockdown of KIF26B significantly diminished the cell migration ability induced by *F. nucleatum* (Fig. 6d, e and Supplementary Fig. 6h, i). In addition, we silenced KIF26B in METTL3-knockdown HCT116 cells (Fig. 6f) and found that

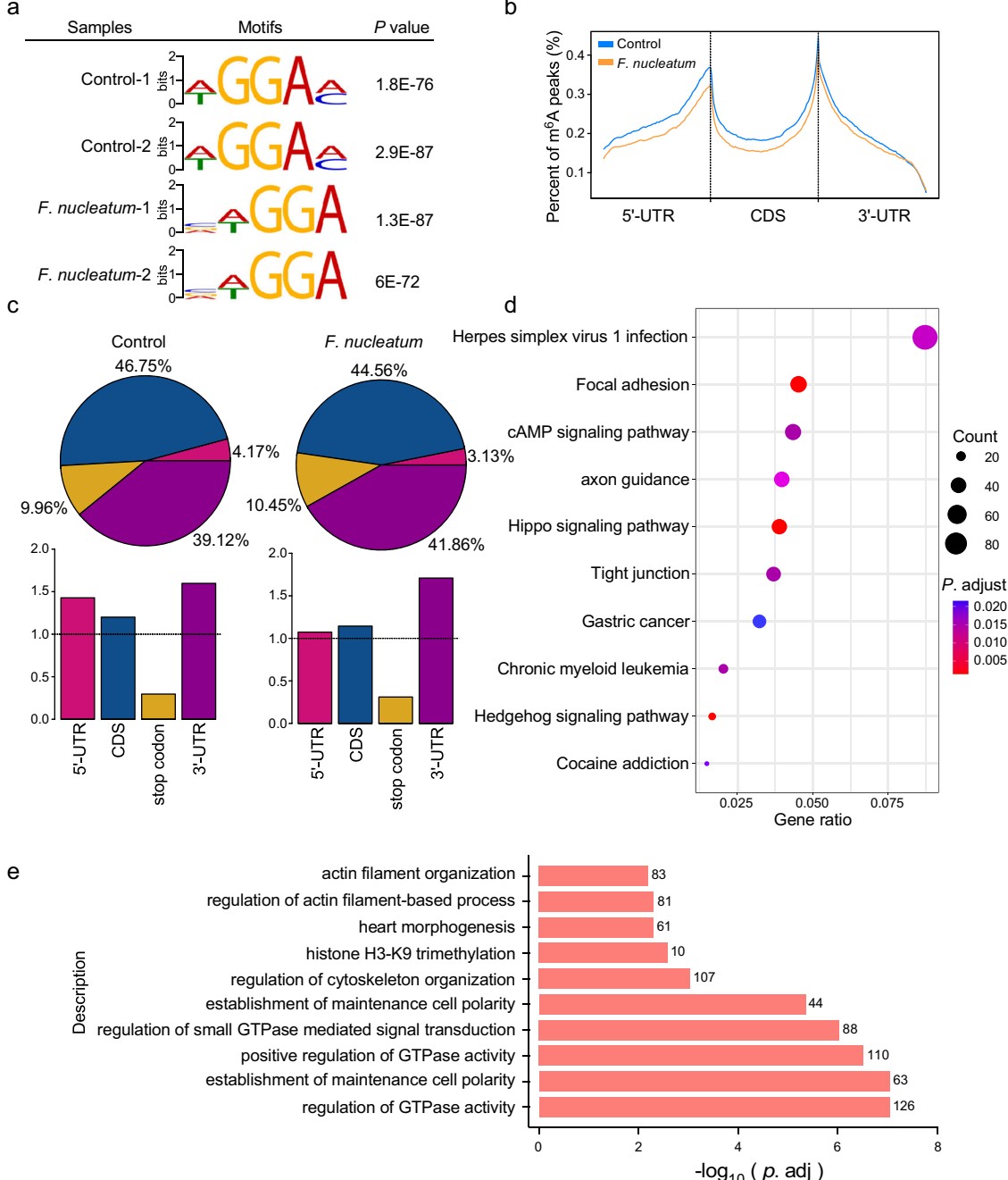

**Fig. 4 Variations of m⁶A-regulated genes in CRC cells with *F. nucleatum* treatment. a** HCT116 cells were treated with *F. nucleatum* or PBS control for 2 h and subjected to m⁶A-sequencing. Predominant consensus motifs identified by DREME within m⁶A peaks of indicated HCT116 cells of two biological replicates with the lowest $p$ value. The significance of the relative enrichment of each motif was computed using the Fisher's Exact Test (*P* value). **b** Density distribution of m⁶A peaks across mRNA transcripts. Regions of the 5′ untranslated region (5′UTR), coding region (CDS), and 3′ untranslated region (3′ UTR) were split into 100 segments, then percentages of m⁶A peaks that fall within each segment were determined. **c** Proportion of m⁶A peak distribution in the 5′UTR, CDS, stop codon, or 3′UTR region across the entire set of mRNA transcripts. **d** KEGG pathway analysis of a total number of 3589 genes with significant differential m⁶A peaks in *F. nucleatum*-treated cells compared with untreated cells. −log₁₀ (*p*.adj) of the ten most enriched pathways are displayed. *P*. adjust, hypergeometric test with Benjamini–Hochberg adjusted. **e** Gene Ontology enrichment analysis of a total number of 3589 genes with differential m⁶A peaks in *F. nucleatum*-treated cells compared with untreated cells. −log₁₀ (*p*.adj) of the ten most enriched gene functions related to the biological processes are displayed. *P*. adjust, hypergeometric test with Benjamini–Hochberg adjusted.

depletion of KIF26B significantly reversed the enhanced cell migration induced by METTL3 knockdown (Fig. 6g), suggesting that KIF26B, at least in partial, mediates the function of METTL3 in CRC aggressiveness.

Another two CRC cells, RKO, and SW620 cells were employed to further verify the essential role of KIF26B in CRC metastasis.

First, the responses of RKO and SW620 cells to *F. nucleatum* were verified in vitro. As expected, *F. nucleatum* treatment down-regulated METTL3 levels and upregulated KIF26B levels in RKO and SW620 cells (Supplementary Fig. 7a, b). Knockdown of KIF26B dramatically inhibited the migration of RKO and SW620 cells (Supplementary Fig. 7c, d) in vitro. Furthermore,

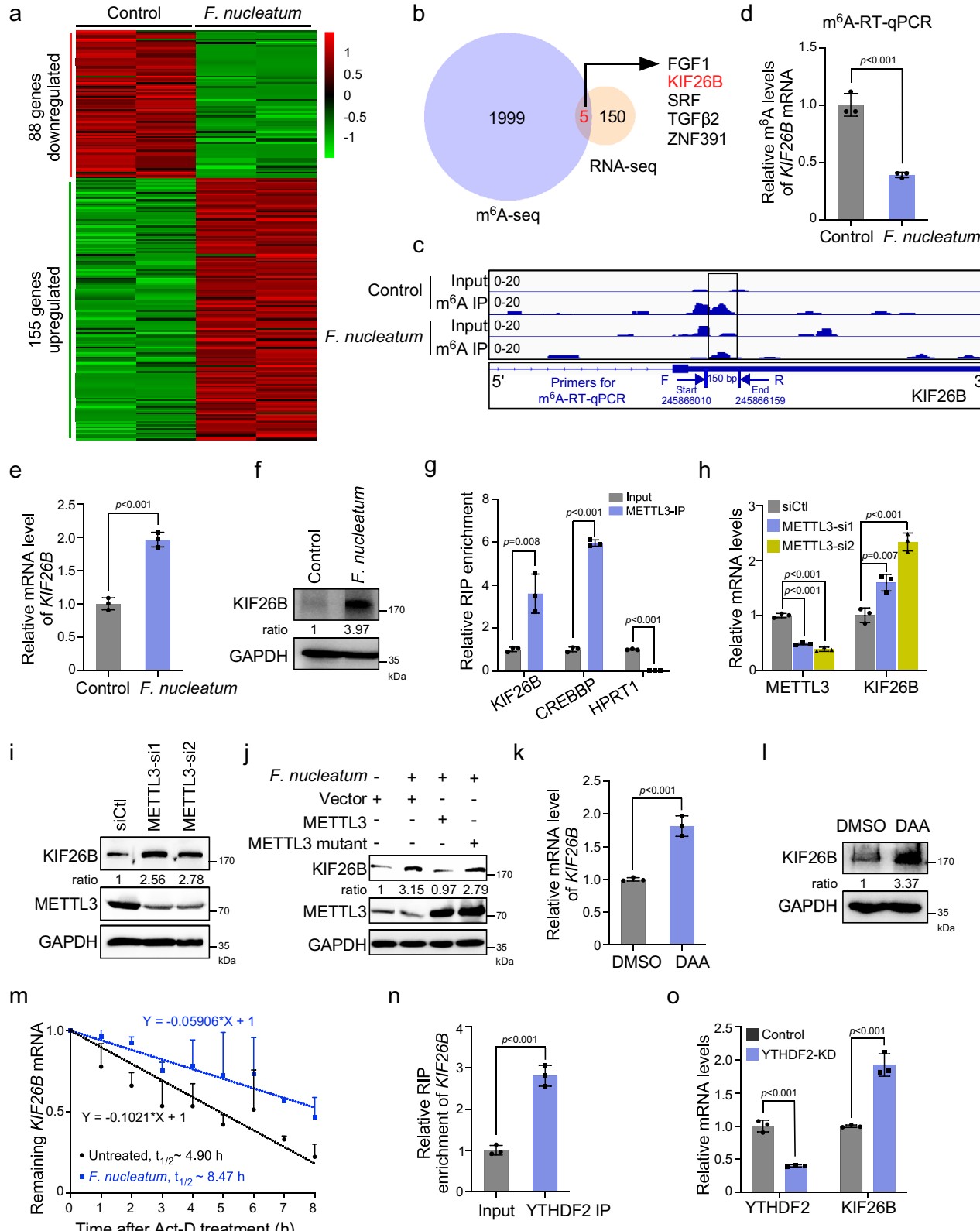

KIF26B-knockdown luciferase-labeled RKO cells or the corresponding control cells were intrasplenically injected into nude mice. Mice who received control CRC cells developed severe metastatic nodules in the liver, as reflected by the gross liver images, bioluminescence imaging, and H&E staining (Fig. 6h). Knockdown of KIF26B robustly reduced the liver metastasis (Fig. 6h, i). Since advanced CRC patients also frequently develop

bone metastasis, we examined bone metastasis in mice and found that knockdown of KIF26B in luciferase-labeled RKO cells clearly inhibited the formation of bone metastatic lesions (Supplementary Fig. 7e, f). Similarly, we also established stable KIF26B-knockdown SW620 cells and performed intrasplenic-injection liver metastasis assays. The mice receiving *F. nucleatum*-treated cells formed obvious liver metastatic nodules compared to those

**Fig. 5 KIF26B is a downstream target of METTL3 by m⁶A-seq and RNA-seq. a** Heat map representing the differential gene expression patterns between *F. nucleatum*-treated and untreated HCT116 cells (fold change >2 or fold change <0.5, P < 0.05). **b** 155 genes with significant mRNA upregulation in RNA-seq and 2004 genes with significant m⁶A peaks downregulation in m⁶A-seq were subjected to Venn diagram analysis. Venn diagram showing the overlapping genes. **c** m⁶A RIP-seq data showed a significant decrease of m⁶A peaks around the stop codon of *KIF26B* mRNA upon *F. nucleatum* treatment. Squares marked a decline of m⁶A peaks in HCT116 cells cocultured with *F. nucleatum*. The region for m⁶A-RT-qPCR detection was annotated. **d** m⁶A RIP-qPCR analysis of *KIF26B* mRNA in the control and *F. nucleatum*-treated HCT116 cells. **e, f** HCT116 cells treated with *F. nucleatum* or PBS control were subjected to quantitative RT-PCR analysis (**e**) and western blot analysis (**f**) of KIF26B expression. **g** RNA Immunoprecipitation (RIP)-qPCR analysis of METTL3 binding with *KIF26B* mRNA, or *CREBBP* mRNA (positive control) or *HPRT1* mRNA (negative control) in HCT116 cells. **h, i** HCT116 cells transfected with indicated siRNAs were subjected to quantitative RT-PCR analysis (**h**) and western blot analysis (**i**) of KIF26B expression. **j** KIF26B expression was detected by western blot in HCT116 cells with indicated treatment. **k, l** Quantitative RT-PCR (**k**) and western blot (**l**) analysis of KIF26B in HCT116 with 10 μM 3-deazaadenosine (DAA) treatment for 48 h. DAA is a methylation inhibitor. **m** HCT116 cells were pretreated with *F. nucleatum* or PBS for 2 h. The remaining *KIF26B* mRNAs were analyzed by quantitative RT-PCR at the indicated time points after actinomycin D treatment. **n** RIP-qPCR analysis of YTHDF2 binding with *KIF26B* mRNA in HCT116 cells. **o** *KIF26B* mRNA levels were detected in YTHDF2-knockdown HCT116 cells. Data were from one representative of three independent experiments (**f, i, j, l, m**). Data were shown as mean ± SD. P values were shown. Two-tailed Student's t-test (**d, e, g, h, k, n, o**).

received untreated or *E. coli*-treated control cells (Fig. 6j, k). Knockdown of KIF26B notably diminished *F. nucleatum*-induced liver metastasis (Fig. 6j, k).

To further validate the role of KIF26B in *F. nucleatum*-induced CRC metastasis, we tried to establish an orthotopic CRC model, following the protocol from previous studies[35,36]. The KIF26B-depleted or control HCT116 cells were pretreated with *E. coli* or *F. nucleatum*, and then orthotopically injected into mice (Supplementary Fig. 7g). H&E staining of whole livers of each mice showed that the mice receiving *F. nucleatum*-treated cells developed more micro-metastases lesions than those received untreated or *E. coli*-treated control cells (Supplementary Fig. 7h, i). Importantly, knockdown of KIF26B notably diminished *F. nucleatum*-induced liver metastasis (Supplementary Fig. 7h, i). Together, our findings indicate that KIF26B is a critical gene for CRC metastasis; *F. nucleatum* promotes CRC aggressiveness and metastasis through upregulation of KIF26B.

**F. nucleatum is correlated with METTL3 and KIF26B expressions in CRC patients.** Recent studies have revealed that a higher abundance of *F. nucleatum* is often detected in CRC tissues[6,7]. To establish a high-*F. nucleatum* environment in the colorectum, C57BL/6 mice after receiving antibiotics streptomycin for 3 days were administered *F. nucleatum* or *E. coli* every day (Fig. 7a). After treatment for 15 days, we observed an enrichment of *F. nucleatum* in the colorectum (Fig. 7b). Notably, the qRT-PCR analysis revealed a significant decrease in *METTL3* mRNA (Fig. 7c) and an increase in *KIF26B* mRNA (Fig. 7d) expression in colorectum tissues from *F. nucleatum*-treated mice compared to the *E. coli* group. Additionally, the decrease in METTL3 protein expression was confirmed by immunohistochemistry staining (Supplementary Fig. 8a). And, we also observed that KIF26B protein was frequently upregulated in the *F. nucleatum*-treated group (Supplementary Fig. 8b). Furthermore, the APC^Min/+ (adenomatous polyposis coli) model was also used to verify our findings. Under the same treatment as the C57BL/6 model. Our data showed that upon exposure to *F. nucleatum* (Supplementary Fig. 8c), the *METTL3* mRNA expression in colorectum tissues significantly reduced compared to that from the *E. coli*-treated mice (Supplementary Fig. 8d), while the mRNA levels of *KIF26B* were significantly increased (Supplementary Fig. 8e). These results provide in vivo evidence that an overabundance of *F. nucleatum* in the mouse intestine leads to downregulation of METTL3 and upregulation of KIF26B.

To determine the clinical relevance of *F. nucleatum*/METTL3/KIF26B in advanced CRC patients, we collected a CRC cohort. Our results showed that the abundance of *F. nucleatum* was significantly higher in tumor tissues than in matched non-tumor

tissues (Fig. 7e). Interestingly, the expression levels of *METTL3* were downregulated in tumor tissues (Fig. 7f), while the *KIF26B* levels were significantly higher in tumor tissues than in matched non-tumor tissue (Fig. 7g). There was a negative correlation between the abundance of *F. nucleatum* and *METTL3* levels in CRC tumor tissues (Fig. 7h), as well as a negative correlation between the expression levels of *METTL3* and *KIF26B* in CRC tumor tissues (Fig. 7i). In addition, we analyzed the relevance of KIF26B expression to patient outcomes. Kaplan–Meier analysis of the Cancer Genome Atlas (TCGA) database showed that high expression of *KIF26B* was significantly associated with a shorter overall survival time of CRC patients (Fig. 7j). Multivariate Cox analysis confirmed that high expression of *KIF26B* was independently associated with the poor prognosis of CRC patients (Fig. 7k).

## Discussion

Gut–microbiota–host interactions have been documented to be involved in CRC progression, as well as in other diseases. However, its underlying mechanisms still remain elusive. Here, our works reveal that *F. nucleatum* treatment induces a dramatic decline of m⁶A modification in both CRC cells and PDX tissues from CRC patients. The decrease in m⁶A levels is due to the downregulation of METTL3 by *F. nucleatum* (Fig. 7l). Mechanistically, our results suggest a microbiota–host interaction model for controlling CRC metastasis. We characterize FOXD3 as a transcription factor for METTL3. *F. nucleatum* treatment activates YAP signaling of CRC cells through inhibiting the Hippo pathway, further reduces the expression of FOXD3, and subsequently inhibits the transcription of METTL3. Downregulation of METTL3 facilitates the expression of target *KIF26B* mRNA by reducing its m⁶A modification levels and diminishing YTHDF2-dependent mRNA degradation. We demonstrate that KIF26B is critical for *F. nucleatum*-induced CRC cell aggressiveness and metastasis.

The microbiome has profound effects on the host physiology by modulating host the transcriptome, proteome or epigenome levels. However, little is known about the regulation of the host RNA epitranscriptome by the microbiome. Very recently, Wang et al. analyzed the m⁶A status of tissues from germ-free (GF) mice and specific-pathogen-free (SPF) mice and found that brain tissues from SPF mice had lower m⁶A levels than the GF mice[16]. Similarly, Jabs et al. reported significant variations in m⁶A modifications in the cecum and liver of SPF mice compared to GF mice, in which multiple signaling pathways, including metabolism, inflammation and antimicrobial responses, were changed[17]. These variations in m⁶A modification can be restored by colonizing mice with a conventional specific-pathogen-free gut

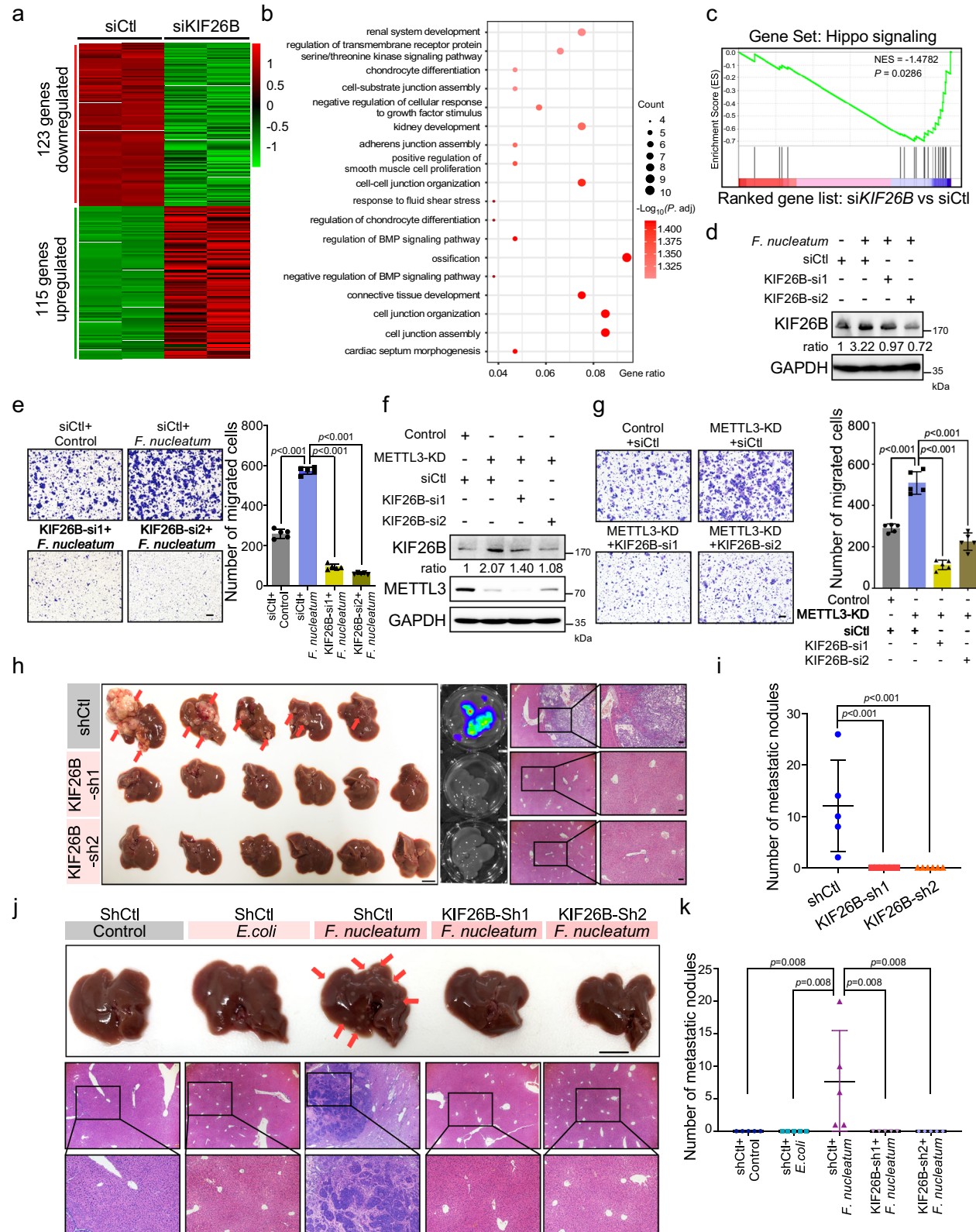

flora[17]. These two studies examined the physiological condition and suggested that the microbiota has a strong influence on the host m6A epitranscriptome in mice. Here, we investigated the effect of microbiota–host interactions on the human m6A epi-transcriptome under pathological conditions and its significance in cancer progression. We treated CRC cells and PDX tissues from CRC patients with *F. nucleatum* and revealed a significant

m6A decrease. Moreover, we described the detailed mechanism by which *F. nucleatum* downregulates METTL3 expression through YAP signaling activation. Additionally, we characterized KIF26B as a target gene of METTL3, which mediated the *F. nucleatum*-induced CRC aggressiveness and metastasis. Our work suggests that microbiome-induced host epitranscriptomic variation has profound effects on host cancer progression.

**Fig. 6 *F. nucleatum* accelerates CRC aggressiveness and metastasis by upregulating KIF26B. a** Hierarchical clustering showing the genes that were differentially expressed in KIF26B-knockdown HCT116 cells and control cells. (fold change >2 or fold change <0.5, *P* < 0.05). **b** Gene Ontology enrichment analysis of the 123 downregulated genes. *P*. adjust, hypergeometric test with Benjamini–Hochberg adjusted. **c** GSEA analyses of KIF26B-regulated gene signature versus Hippo signaling signature. *P* value was analyzed by a hypergeometric test. **d, e** HCT116 cells transfected with two siRNAs targeting KIF26B or control siRNAs were treated with *F. nucleatum* for 2 h. Western blot analysis (**d**) and transwell migration analysis (**e**) were performed. The migrated cells were quantified in five fields (Right). Scale bar, 100 μm. **f** Western analysis of KIF26B in HCT116 cells transfected with the indicated siRNAs. **g** The HCT116 cells with indicated treatments were applied for transwell migration analysis. Representative images of migrated cells were shown (Left). The migrated cells were quantified in five fields (Right). Scale bar, 100 μm. **h, i** Luciferase-labeled RKO cells were stably infected with lentivirus-based KIF26B shRNAs (*n* = 6) or control shRNAs (*n* = 5). The indicated cells were intrasplenically injected into nude mice to develop liver metastasis. Representative gross livers (left) (Scale bar, 1 cm), bioluminescence images (middle), and H&E stained liver sections (right, Scale bar, 20 μm) of the mice are shown (**h**). Liver metastatic nodules per mice were quantified (**i**). **j, k** SW620 cells were stably infected with lentivirus-based KIF26B shRNAs or control shRNAs. The indicated cells were cocultured with *E. coli* or *F. nucleatum* for 24 h and intrasplenically injected into nude mice. Representative gross livers (Scale bar, 1 cm) and H&E stained liver sections (Scale bar, 20 μm) of the mice are shown (**j**). Liver metastatic nodules per mice were quantified (**k**) (*n* = 5). Data were from one representative of three independent experiments (**d, f**). Data were shown as mean ± SD. *P* values were shown. Two-tailed Student's *t*-test (**e, g**). Two-tailed Mann–Whitney test (**i, k**).

METTL3 has been found to play diverse biological functions in many cancer types[37,38], and most of the studies focused on its downstream target genes. However, the upstream mechanisms for regulating METTL3 have been rarely explored. Wang et al. suggested that the high expression of METTL3 in gastric cancer was mediated by chromatin acetylation[39]. Our results suggest FOXD3 as a transcription factor inducing METTL3 expression by binding to the promoter regions of METTL3. FOXD3, a member of the forkhead box family, has been suggested to be a tumor-associated transcription factor. In addition, our findings reveal that microbiota–host interactions can activate YAP signaling of CRC cells. Our results showed that *F. nucleatum* treatment obviously inhibited the phosphorylation of MST, as well as its upstream factors NF2, KIBRA, and WILLIN\FRMD6, suggesting that *F. nucleatum* may inactivate the HIPPO signaling pathway. However, the exact upstream mechanisms of the HIPPO pathway, such as how the extracellular events stimulate NF2, KIBRA, WILLIN\FRMD6, and MST complex, is still a mystery in the HIPPO-YAP field. Some secreted proteins or metabolites produced by *F. nucleatum* may be involved in the process. It will be meaningful to investigate how *F. nucleatum* suppresses the Hippo pathway and explore the other upstream mechanisms for modulating METTL3 in the future.

Since METTL3 is the most important methyltransferase for mRNA m6A modifications which play diverse functions in many biology processes and diseases, it is possible that METTL3 acts either oncogenic or tumor-suppressive roles in different stages of CRC progression by modulating different target pathways, as the previous reports[37,38,40]. Here, we revealed that the gut microbes *F. nucleatum* can reduce the m6A levels of CRC cells through inhibition of METTL3, leading to enhanced CRC cell migration and metastasis. Under the condition of *F. nucleatum* treatment, METTL3 acted as a tumor suppressor gene for CRC metastasis by downregulation of target KIF26B expression. However, the diverse functions and mechanisms of METTL3 in CRC progression, as well as other cancers, need further studies.

KIF26B, a member of the kinesin superfamily, is involved in the regulation of intracellular transport. Recently, KIF26B has been reported to promote gastric cancer, hepatocellular carcinoma, and breast cancer progression[33,34,41]. However, the effects and potential mechanisms of KIF26B in CRC metastasis are still unclear. Here, we found that KIF26B, at least in partial, mediates the functions of *F. nucleatum* in promoting the cell migration and metastasis of CRC cells. High KIF26B expression is significantly associated with a shorter survival time of CRC patients. In addition, although we identified KIF26B as a direct target of METTL3 under *F. nucleatum* treatment and suggested that *KIF26B* mRNA is regulated by YTHDF2-dependent mRNA

degradation, we cannot rule out the possibility that there are many other targets of METTL3 under *F. nucleatum* treatment. Integration of RNA-seq and m6A-seq could not appropriately reflect the effect of m6A modification on mRNA translation. It will be interesting to identify other targets of METTL3 in microbiota–host interactions by combining these results with ribosome-seq or proteomic datasets. Moreover, the role of *F. nucleatum* in CRC cell proliferation has been well reported[3,10]. It will be interesting to explore whether METTL3 is involved in *F. nucleatum*-induced cell proliferation and its underlying mechanism.

In summary, our study provides critical insights into the effect of the gut–microbiota on the human m6A epitranscriptome and its role in promoting CRC metastasis. *F. nucleatum* induces a reduction in m6A modification through the YAP/FOXD3/METTL3 axis, resulting in the upregulation of KIF26B and enhanced aggressiveness of CRC cells. A potential therapeutic strategy targeting either *F. nucleatum* or KIF26B could be well utilized. Furthermore, it should be noted that additional epitranscriptomic mechanisms are likely to exist in other patterns of intestinal microbiota–host interactions, which needs further investigation to expand the horizon on microbiota-directed regulation.

## Methods

**Human specimens**. All samples were obtained with informed consent from previously untreated CRC patients from Sir Run Run Shaw Hospital, Zhejiang University School of Medicine (Hangzhou, China). Ethical consent was approved by the Institutional Review Board of Sir Run Run Shaw Hospital.

**Patient-derived xenografts (PDX)**. Surgical specimen from colorectal cancer patient was physically separated into small pieces (2–3 mm) and subcutaneously transplanted into 5- to 6-week-old female nude mice. Once tumors were grown, xenografts were resected, and cut into similarly sized PDX tissues. The obtained PDX tissues were divided into three groups and treated with *F. nucleatum*, *E.coli* DH5α or PBS separately at the MOI of 100:1 for 24 h. After washing with PBS, total RNAs were extracted from the PDX tumor tissues by Trizol reagent (Invitrogen, USA) following the manufacturer's instructions for subsequent experiments.

**Bacterial strains and growth conditions**. The *F. nucleatum* strain was purchased from the American type culture collection (ATCC, *F. nucleatum* strain 25586). *F. nucleatum* was grown in Columbia blood agar supplemented with 5 μg/ml haemin, 5% defibrinated sheep blood, and 1 μg/ml vitamin K1 (Sigma-Aldrich, St. Louis., MO, USA) in an anaerobic jar (MITSUBISHI Gas Chemical Co., Japan) at 37 °C. The *E. coli* strain DH5α (Takara, Japan) was cultured in Luria-Bertani (LB) agar plate at 37 °C. According to previous studies, *E. coli* and *F. nucleatum* was used at MOI of 100:1[3,11,26].

**Cell culture**. The CRC cell lines (HCT116, LoVo, RKO, SW620) were provided by the American Type Culture Collection (ATCC, Manassas, VA, USA). The luciferase-labeled RKO cell line was donated by Maode Lai's lab[42]. HCT116 cultured in Maccoy 5 A (Genom, China), LoVo cultured in F-12K (Genom, China),

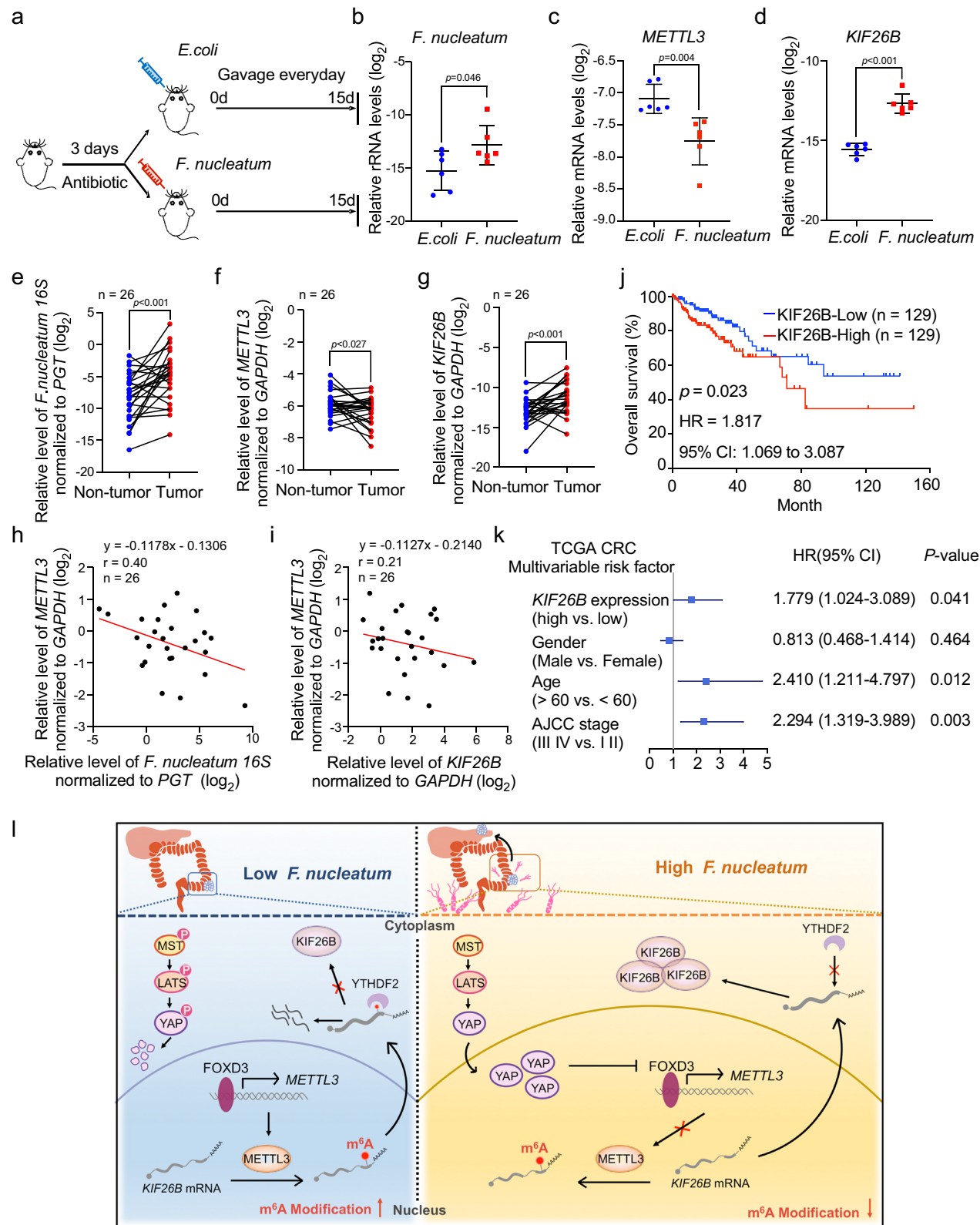

RKO cultured in high-glucose DMEM (Genom, China), and SW620 cultured in 1640 (Genom, China) were supplemented with 10% fetal bovine serum (FBS). All of the cell lines were cultured at 37 °C in humidified 5% $CO_2$ atmosphere. All cell lines we used in this study were mycoplasma free.

**m6A dot blot assay**. CRC cells were cocultured with *E. coli* or *F. nucleatum* for the indicated time (2 or 24 h). PDX tissues were cocultured with or without *F. nucleatum* for 24 h. The total RNAs were isolated using Trizol reagent (Invitrogen,

USA). RNA dot blot analysis was performed as previously described[43]. Briefly, we used the magnetic messenger RNA insolated kit (NEB, USA) to obtain poly (A)+ mRNA. The indicated amount of mRNA was spotted onto Hybond N+ membranes (GE healthcare) and using m6A specific antibody (Synaptic Systems, Germany) for dot blot analysis.

**UHPLC Q-exactive MS analysis of RNA**. A total of 200 ng of mRNA were digested by Nuclease P1 (NEB, USA) in a 25 μl reaction system at 37 °C for 1 h,

**Fig. 7 *F. nucleatum* is correlated with METTL3 and KIF26B expressions in CRC patients. a** C57BL/6 mice pretreated with 3-day antibiotics were administrated with *F. nucleatum* or *E. coli* everyday by gavage. Mice were sacrificed after the treatment for 15 days. **b** Quantitative RT-PCR analysis of *F. nucleatum* in colorectum tissues from the indicated mice ($n = 6$). **c, d** Quantitative RT-PCR analysis of *METTL3* mRNA (**c**) and *KIF26B* mRNA (**d**) expression in colorectum tissues from the indicated mice ($n = 6$). **e** Quantitative RT-PCR analysis of *F. nucleatum* in human CRC tissues and matched adjacent non-tumor tissues. The CRC patients in this cohort all had lymph node metastasis ($n = 26$). **f, g** Quantitative RT-PCR analysis of *METTL3* (**f**) and *KIF26B* (**g**) in human CRC tissues and matched adjacent non-tumor tissues. **h, i** The correlation between the relative levels of *F. nucleatum 16 S* and *METTL3* mRNA (**h**) or the correlation between the levels of *METTL3* and *KIF26B* mRNA (**i**) in CRC tissues and adjacent non-tumor tissues ($n = 26$). The correlation was analyzed by linear regression. **j** Kaplan–Meier survival curves were analyzed and compared between patients with low ($n = 129$) and high ($n = 129$) levels of *KIF26B* in CRC patients from The Cancer Genome Atlas (TCGA) database. Subgroups with high- or low-*KIF26B* expression were sorted according to the TPM gene expression standard values. **k** Multivariable analysis was performed of CRC patients in the TCGA database. The center dots represent the Hazard ratio (HR). All the bars correspond to 95% confidence intervals. *P* value, multivariate cox regression analysis. The details were provided in the methods section. **l** Schematic diagram of the *F. nucleatum*-induced downregulation of METTL3-mediated m$^6$A modifications contributes to CRC metastasis. Data of quantitative RT-PCR are presented as log$_2$ value normalized to universal *Eubacteria 16 S* (**b**), *GAPDH* (**c, d, f, g**), or *PGT* (**e**). Data were shown as mean ± SD. *P* values were shown. Two-tailed Mann–Whitney test (**b–d**), Two-tailed paired *t*-test (**e–g**), and log-rank test (**j**). AJCC American Joint Committee on Cancer, CI confidence interval, HR hazard rate.

which was followed by the treatment of Antarctic Phosphatase (NEB, USA) and additional incubation at 37 °C for 2 h. Samples were then diluted to 105 μl and centrifuged at the speed of $15,000 \times g$ for 10 min. Ten microliters of the solution was loaded into UHPLC Q-Exactive MS (Q-Exactive High-Resolution Benchtop Quadrupole Orbitrap Mass Spectrometer). The samples were run in mobile phase buffer A (water with 0.1% Formic Acid) and 2 to 98% gradient of buffer B (Methanol with 0.1% Formic Acid). Nucleosides were quantified by using retention time and nucleoside to base ion mass transitions of 282.1197 to 150.0774 (m$^6$A) and 268.1040 to 136.0618 (A).

**Migration and invasion assay.** CRC cell lines were incubated with PBS, *E. coli*, or *F. nucleatum* for 2 h in advance. For migration assay, the $2.5 \times 10^5$ cells were suspended in 100 μl medium with 1% serum were seeded in the upper chamber of transwell chambers (8 μm pores, Corning, USA), and 800 μl fresh medium with 10% serum were added to the lower chamber. For invasion assay, 100 ul diluted Matrigel (Matrigel:PBS = 1:9) was added in the upper chamber of transwell chambers (8 μm pores, Corning, USA) for 2 h. Before seeding the cells, the Matrigel was discarded. After incubation for 24 h at 37 °C, the cells in the upper chamber were fixed with 4% paraformaldehyde, followed by staining with 0.1% crystal violet. The migrated cells were quantified by counting in five fields (100×) under an Olympus IX2-ILL100 optical microscope (Olympus, Center Valley, USA). Each experiment was analyzed in triplicate.

**Plasmid construction.** To construct reporter plasmids containing METTL3 promoter, sequence −2000 bp ~100 bp of METTL3 promoter region were constructed into the pGL3-basic vector (Promega # E1751). To generate shRNA constructs of KIF26B and FOXD3, oligos targeting mRNA of KIF26B or FOXD3 were synthesized and cloned into lentiviral pLKO.1 puro vector (Addgene plasmid # 8453). The shRNA sequences we used were listed in the following table. The full-length OFR of human METTL3 (NM_019852.5) and FOXD3 (NM_012183.3) were prepared by cloning the cDNAs from HEK293 cells into the pcDNA3.1(+) vector (Invitrogen # V790-20). The full-length OFR of human FOXD3 (NM_012183.3) were also cloned into pcDNA3.1(+)/myc-His C vector (Invitrogen # V80020). The primers were listed in Supplementary Table 3. The pcDNA3.1(+)-YAP vector was donated by Song's lab.[44] The pGL4-basic-CTGF vector was donated by Zhao's lab[27].

**Lentivirus production, precipitation, and infection.** To generate cells stably expressing shRNAs, HEK293T cells were transfected with the indicated lentivirus expression vector and viral packaging constructs. Forty-eight hours after transfection, the viral medium was collected, and concentrated overnight by Lentivirus Concentration Kit (Genomeditech, China). The lentivirus was centrifuged at $4000 \times g$ at 4 °C for 25 min, and the resuspended precipitation was used to infect the target cells. After 24 h of infection, cells were treated with puromycin (MCE, USA) for 4–5 days.

**RNA extraction and qRT-PCR.** Total RNA was extracted from CRC cell lines or fresh frozen samples using Trizol reagent (Invitrogen, USA), and cDNA was reversed by PrimeScript RT Reagent Kit (Takara, Japan) according to the manufacturer's protocol and stored at −20 °C. Quantitative real-time PCR was performed in triplicates in ROCHE LightCycler®480 System (Rotor gene 6000 Software, Sydney, Australia). GAPDH was served as an internal reference gene. The primers are shown in Supplementary Table 1.

**DNA extraction and *F. nucleatum* quantification.** Genomic DNAs from clinical and mice samples were extracted using QIAamp DNA Mini Kit (QIAGEN, Germany). *F. nucleatum* quantification was performed by quantitative real-time PCR

as described above. Universal *Eubacteria 16 S* was used as a reference control. The primers are shown in Supplementary Table 1.

**Western blot analysis.** Cells extracts were collected using Radio-Immunoprecipitation Assay (RIPA) buffer containing protease inhibitor (Roche, Switzerland). Total protein was electrophoresed through 10% SDS polyacrylamide gels and then transferred to PVDF membranes. The membranes were blocked in 5% fat-free milk for 2 h and then incubated with the indicated primary and secondary antibodies. The information of antibodies was shown in Supplementary Table 4. The signal was detected using an ECL kit (Fdbio science, China) by ChemiDoc Touch Imaging System (Bio-Rad, USA).

**m$^6$A sequencing.** For m$^6$A sequencing, HCT116 cells were treated with PBS or *F. nucleatum* (MOI of 100:1) for 2 h. Total polyadenylated RNA was isolated from HCT116 cells using Trizol reagent (Invitrogen, USA) followed by isolation through NEBNext Poly(A) mRNA Magnetic Isolation Kit (New England Biolabs, UK). Then fragmented mRNA to 60–200 bp mRNA fragments were incubated with Dynabeads Protein A (Thermo Fisher Scientific, USA) or m$^6$A specific antibody (Synaptic Systems, Germany), and collected the mRNA fragments with m$^6$A modifications. The library preparation was performed after finishing reverse transcription, second-strand synthesis, end repair, adenylate 3′ ends, and fragments screening test. Illumina Hiseq platform was used to sequence the library. Each experiment was conducted with two biological replicates. The adapters were removed by using cutadapt for m$^6$A-seq, reads were aligned to the reference genome (hg19) in HISAT v2.1.0. Significant peaks were analysed by MACS2 using Fisher exact test (Threshold value = 1, FDR < 0.05). Gene expression was calculated by stringTie using sequencing reads from input samples. Differential expression genes were analyzed using DEseq2. The identified genes with differential peaks were used for subsequent GO and KEGG analysis. GO and KEGG were analyzed using hypergeometric test adjusted with Benjamini–Hochberg method (P. adj). For the analysis of sequence consensus, DREME was used as previously reported[45]. The significance of the relative enrichment of each motif was computed using the Fisher's Exact Test (P value).

**RNA sequencing.** For RNA sequencing, total RNA was isolated from indicated cells using Trizol reagent (Invitrogen, USA). mRNA was purified from total RNA using poly-T oligo-attached magnetic beads. Sequencing libraries were generated using NEBNext® UltraTM RNA Library Prep Kit for Illumina® (NEB, USA) following the manufacturer's recommendations. About 150 bp paired-end libraries were sequenced by the Illumina PE150 platform. Paired-end clean reads were aligned to the human genome version hg19 using HISAT2 v2.0.5. Differential expression analysis of two groups was performed using the DESeq2. P value < 0.05 and |log$_2$ (fold change)| >1 were considered as significant.

**Luciferase reporter assay.** For METTL3-promoter activity assay, the upstream −2000 bp of a promoter of METTL3 was constructed into a pGL3-basic vector (Promega # E1751). CRC cells were transfected with pcDNA3.1(+)/myc-His C vector (Invitrogen # V80020) control or pcDNA3.1(+)/myc-His C-FOXD3 plasmid, and co-transfected with the reporter plasmids (pGL3-basic-METTL3 (Promoter) vector and TK-renilla vector). For CTGF-promoter activity assay, CRC cells were transfected with the reporter plasmids (pGL4-basic-CTGF[27] and TK-renilla vector). The transfected cells were treated with or without *F. nucleatum* (MOI of 100:1) for 2 h. Luciferase assays were measured at 24 h after transfection, using Dual-Luciferase Reporter System (Promega, Fitchburg, WI, USA) according to the manufacturer's protocol. The transfection efficiency data were presented by normalizing the Firefly luciferase activities to that of Renilla luciferase signals. Each assay was performed independently in triplicate.

**Animal assays**. All mice used in this study were approved by the Institutional Animal Care and Use Committee of Zhejiang University. BALB/c Nude mice (5–6 weeks old) and C57BL/6 mice (5–6 weeks old) were used for indicated studies, kept in SPF facilities (25 °C, suitable humidity (typically 50%), 12 h dark/light cycle). KIF26B-knockdown RKO-Luc cells ($1 \times 10^6$ cells/per mouse) were intrasplenically injected into nude mice for developing liver metastasis. For the bone metastasis model, KIF26B-knockdown RKO-Luc cells ($1 \times 10^6$ cells/per mouse) were intravenously injected into the nude mouse. After 2 months, the mice were sacrificed and the Xenogen IVIS 200 Imaging System was used to measure the development of liver metastases and bone metastases. For SW620 cells liver metastasis model, SW620 cells or KIF26B-knockdown SW620 cells were incubated with *E. coli* or *F. nucleatum* (MOI of 100:1) for 24 h and intrasplenically injected into nude mice ($1 \times 10^6$ cells/per mouse). After 2 months, the mice were sacrificed. Stereo microscope image (Olympus SZX16, Japan) and H&E staining of liver tissue sections were used to evaluate tumor metastasis.

For the orthotopic model, HCT116 cells or KIF26B-knockdown HCT116 cells were incubated with *E. coli* or *F. nucleatum* (MOI of 100:1) for 24 h and orthotopically injected into NOD SCID mice ($2 \times 10^6$ cells in 50 ul/per mouse, PBS: Matrigel = 1:1). The next day, live imaging was used to confirm whether modeling successful. Forty-five days after modeling, the mice were sacrificed. The colon of the mice was dissected and H&E staining of liver tissue sections were used to evaluate tumor metastasis. For *F. nucleatum*-treatment C57BL/6 mice and APC$^{Min/+}$ mice model, C57BL/6 mice (5–6 weeks old) or APC$^{Min/+}$ mice (2-month old) pretreated with 2 mg/ml streptomycin by gavage administration for 3 days. Then, the mice were administrated with $10^9$ CFU *F. nucleatum* or *E. coli* every day. After 15 days, mice were sacrificed and the colorectum of mice were surgically excised for further analysis.

**Immunofluorescence**. CRC cells were seeded on chamber slides until adhered and cocultured with *F. nucleatum* (MOI of 100:1) for 2 h. After washing with PBS, the cells were fixed with 4% paraformaldehyde for 15 min at room temperature (RT). The slides were incubated with 0.2% Triton X-100 for 15 min and blocked in 3% BSA for 30 min. After washing by PBST three times, the cells were incubated with primary antibody against YAP (Proteintech, China) at RT for 2 h. Then the slides were washed by PBST three times and incubated with secondary antibodies and DAPI at RT for 1 h. The cells were fixed with Gold Antifade Reagent (Invitrogen, USA) and imaged using LSM 800 with Airyscan confocal laser-scanning microscope (Zeiss, Germany).

**RNA binding protein immunoprecipitation (RIP)-qPCR**. The CRC cells were washed twice with ice-cold PBS and lysed in 1 ml RIP Lysis buffer (150 mM KCl, 25 mM Tris, 5 mM EDTA, 0.5% Triton X-100, 0.5 mM DTT, Protease inhibitor (1:100), RNAase inhibitor (1:1000)) on ice for 30 min. Cell lysates were centrifuged at $12,000 \times g$ at 4 °C for 15 min. 10% supernatant was collected as input, and the remaining supernatant was incubated with anti-METTL3 (anti-YTHDF2) antibody or control IgG antibody that protein A/G conjugated magnetic beads (MCE, USA) in 900 μl RIP Lysis buffer at 4 °C for 4 h. Bound RNAs were immunoprecipitated with beads and wash beads with RIP buffer (150 mM KCl, 25 mM Tris, 5 mM EDTA, 0.5% Triton X-100) four times. Then treated the beads with 10 μl 10% SDS, 10 μl Proteinase K, and 130 μl RIP buffer for 30 min at 55 °C. RNA in IP or Input group was recovered with Trizol reagent (Invitrogen, USA) according to the manufacturer's instruction and analyzed by quantitative RT-PCR. IP enrichment ratio was calculated as a ratio of its amount in IP to that in the input.

**m⁶A-RT-qPCR**. Total RNA was extracted from CRC cell lines using Trizol reagent (Invitrogen, USA). About 100 ug of total RNA were digested by DNase (Takara, Japan) in a 150 ul reaction system at 37 °C for 20 min. After digestion, the total RNA was extracted again using Trizol reagent, which was followed by RNA fragmentation using fragmentation reagents (Invitrogen, USA) at 71 °C for 5 min. Then the stop buffer was added immediately, vortexed, and placed the RNA on ice. The fragment RNA was extracted using Trizol reagent, and dissolved in 200 ul DEPC water. About 160 ul of fragmented RNA was diluted with Me-RIP buffer (150 mM KCl, 25 mM Tris, 5 mM EDTA, 0.5% Triton X-100, 0.5 mM DTT, Protease inhibitor (1:100) (Invitrogen, USA), RNAase inhibitor (1:1000) (ABclonal, China)) and divided into two tubes, which was incubated with anti-m⁶A (ABclonal, China) antibody or control IgG antibody that protein A/G conjugated magnetic beads (MCE, USA) in 900 μl RIP Lysis buffer at 4 °C for 4 h. 20% of fragmented RNA was collected as input. Bound RNAs were immunoprecipitated with beads and wash beads with RIP buffer four times. Then treated the beads with 10 μl 10% SDS, 10 μl Proteinase K (Takara, Japan), and 130 μl Me-RIP buffer for 30 min at 55 °C. Transfer the liquid to new tubes and 1 ml Trizol reagent was added to each tube. Chloroform was added and after centrifuging, the upper water phase was taken. 1/10 volume of 3 M sodium acetate, an equal volume of isopropyl alcohol, and glycogen with a final concentration of 100 ug/ml was added. Keep the samples at −80 °C overnight, then centrifuged at $12,000 \times g$ at 4 °C for 15 min, and washed twice with 75% ethanol. The precipitation was dissolved with equal volume DEPC water and analyzed by One-Step quantitative RT-PCR (TransGen, China).

**Chromatin immunoprecipitation assay (ChIP)**. HCT116 cells were transfected with pcDNA3.1(+)/myc-His C control vector or pcDNA3.1(+)/myc-His C-FOXD3 vector for 48 h. Then the ChIP assay was performed using the Abcam ChIP kit (Abcam, Cambridge, MA) according to the manufacturer's instruction. The cells were washed twice with ice-cold PBS and collected. 1 ml 1% formaldehyde were added and incubated for 10 min at room temperature. Then the reaction was stopped by the addition of 125 mM glycine. Cells were washed with ice-cold PBS, then collected by centrifugation at $300 \times g$ at 4°C, and resuspended in 1 ml of ChIP sonication buffer. Shear DNA using a sonicator to an optimal DNA fragment size and confirmed by agarose gel electrophoresis. Pellet cell debris by centrifugation, $14,000 \times g$, 15 min, 4 °C. 10% supernatant was collected as input, and the remaining supernatant was incubated with an anti-MYC-tag antibody at 4 °C overnight. Protein A/G Magnetic Beads were added to the supernatant and incubate on a rotary homogenizer at 4 °C for 4 h. The immunoprecipitated DNA fragments were extracted by using a PCR purification kit (Qiagen, German) and analyzed by quantitative RT-PCR. The primers for detecting FOXD3 binding to the *METTL3* promoter region are listed in Supplemental Table 1.

**mRNA stability**. *F. nucleatum*-pretreated cells and the corresponding control cells were treated with actinomycin D (MCE, China, 10 μg/ml) to inhibit further RNA synthesis. Cells were harvested at the indicated time points, total RNAs were extracted. The remaining mRNA levels of KIF26B in each group were detected by quantitative RT-PCR. The remaining RNA levels at each time point were normalized to that of the first time point (0 h).

**Transcription factors prediction**. The online bioinformatics tools *AnimalTFDBI* (http://bioinfo.life.hust.edu.cn/AnimalTFDB/) (Binding score top1000; *P* value top1000) and *TRANSFAC* (http://gene-regulation.com/pub/databases.html) were used to scan the potential transcription factors of METTL3. The sites with the highest binding score for each transcription factor were reserved. JASPAR (http://jaspar.genereg.net/) was used to evaluate the binding score of two selected transcription factors.

**Bioinformatics analysis**. A total number of 380 colorectal cancer patients from the TCGA database with TPM gene expression standard values and complete pathological information were available for analysis. The patients were ranked according to the expression levels of *KIF26B*. The top 1/3 patients were defined as the KIF26B high expression group ($n = 129$), and the bottom 1/3 patients were defined as the KIF26B low expression group ($n = 129$). These two groups of patients were applied for Kaplan–Meier survival curves analysis and multivariable analysis.

**Statistics and reproducibility**. Each experiment was performed at least three biological replicates. Results are presented as the mean ± SD, comparisons were made using two-tailed Student's *t*-test, two-tailed paired t-test or two-tailed non-parametric Mann–Whitney *U*-test, and $p < 0.05$ indicates statistical significance. Statistical analyses were performed using the GraphPad Prism 7 software. Liver and bone metastases were monitored by BLI and quantified by Living Image software. The dot blot and western blot results were quantified using Image J 1.53 software and normalized to the loading control or GAPDH, respectively.

**Reporting summary**. Further information on research design is available in the Nature Research Reporting Summary linked to this article.

## Data availability

The accession number for the data for the m⁶A-seq reported in this study is NCBI GEO: GSE150308. The accession number for the data for RNA-seq reported in this study is NCBI GEO: GSE150309. All the other data supporting the findings of this study are available within the article and its Supplementary Information files. A reporting summary for this article is available as a Supplementary Information file. Source data are provided with this paper.

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

## Acknowledgements

This work was supported by the National Natural Science Foundation of China (81972276 to W.Z., 82173040 to W.Z., 82022623 to L.J.W., 31972883 to C.H.H., 31771540 to W.Z., and 91740205 to T.H.Z.), Natural Scientific Foundation of Zhejiang Province, China (LYY19H310011 to B.Y.W.). We thank Maode Lai for providing the luciferase-labeled RKO CRC cell lines, Hai Song for providing the pcDNA3.1(+)-YAP vector, and Bin Zhao for providing the pGL4-CTGF vector.

## Author contributions

L.Z., M.J.L., Y.Z., and W.Z. designed the experiments and interpreted the results. L.Z., M.J.L., Y.Z., M.S., L.F.W., J.B.L., Y.C., Q.C., C.Q.J., and X.L. performed the experiments. S.J.C., L.J.W., and B.Y.W. provided the patient samples. L.J.W., C.H.H., H.C., T.H.Z., and S.J.C. improved the project design and interpreted the results. L.Z., W.Z., and Y.Z. wrote the manuscript, which was further refined by all authors. W.Z. supervised the overall study.

## Competing interests

The authors declare no competing interests.
