## [Peer Review File · Nature Communications]

Fusobacterium nucleatum reduces METTL3-mediated m⁶A modification and contributes to colorectal cancer metastasisREVIEWER COMMENTS

Reviewer #1 (Remarks to the Author):

In this manuscript, Chen and colleagues investigated the role of METTL3 and m6A RNA methylation in colorectal cancer (CRC) metastasis promoted by the pathogenic bacterium *Fusobacterium nucleatum* (*F. nucleatum*). They found that *F. nucleatum* down-regulates the transcription of METTL3, as well as m6A RNA methylation, through activating YAP signaling and subsequently inhibiting FOXD3 activity. Down-regulation of METTL3 increases the mRNA stability of KIF26B, a gene that is critical for CRC metastasis. This manuscript presented novel insights into CRC metastasis under the influence of *F. nucleatum*. However, the main findings of the manuscript is somewhat observational and correlative. Some of the main conclusions are not supported, including (1) the functional significance of METTL3 down-regulation by *F. nucleatum* in CRC aggressiveness, (2) the mechanism by which *F. nucleatum* regulates the YAP signaling, (3) the mechanism by which KIF26B regulates cell migration and CRC metastasis. The following concerns need to be addressed.

Specific comments:

1. One major weakness of the manuscript is the lack of evidence to support the importance of METTL3 down-regulation in CRC aggressiveness, which limits the impact of the manuscript. For example, does knockdown or overexpression of METTL3 affect CRC aggressiveness with or without *F. nucleatum* treatment? Does KIF26B knockdown modulate the effect of METTL3 knockdown/overexpression in CRC aggressiveness with or without the treatment of *F. nucleatum*?
2. Fig. 1e and extended Fig. 1c: The authors stated that “*F. nucleatum* treatment significantly reduced the expression of METTL3...”. Statistical analysis and SD or SE need to be shown to support this conclusion.
3. Fig. 2a, b: NF- κ B pathway is activated by *F. nucleatum*. Later in extended Data Fig. 2c, the authors showed that knockdown of p65 had not effect on METTL3 expression. However, this is under homeostatis condition, not with the treatment with *F. nucleatum*. They need to test whether NF- κ B inhibition has a role in *F. nucleatum*-induced METTL3 down-regulation.
4. Fig. 3e: the effect of FOXD3 on METTL3-luciferase activity is modest.
5. Fig. 3I: this figure panel showed that the decreased FOXD3 expression is not important for *F. nucleatum*-induced reduction of METTL3 expression. What is the mechanism then? How does *F. nucleatum* inhibit the interaction of FOXD3 with the METTL3 promoter?
6. Fig. 5c: what are the peaks before the stop codon?
7. Fig. 5F: What criteria is used to justify “strong interaction between KIF26B mRNA and METTL3..”.
8. Fig. 5n,o: does YTHDF1 knockdown affect the mRNAs stability of KIF26B with or without *F. nucleatum* treatment? Knockdown of YTHDF1-3 seems to have similar effect on KIF26B levels to YTHDF2 knockdown. How about knockdown of each YTHDF protein? Does knockdown of YTHDF1 or YTHDF3 affect the mRNA stability of KIF26B?
9. How does *F. nucleatum* activate YAP signaling?
10. Fig. 2h, 2i, 3b, 3c, 3f, 3i: the m6A levels need to be determined when the upstream signaling for METTL3 expression is manipulated to support the importance of these METTL3 regulators in controlling m6A enrichment.

Reviewer #2 (Remarks to the Author):

In this study, the authors found that colorectal cancer enriched *Fusobacterium nucleatum*, a well-known oncogenic bacterium, suppressed m6A levels in CRC cell lines and CRC patient cohort. It was further indicated that *F. nucleatum*, through the inhibition of the Hippo pathway, causes activation of YAP. YAP activation inhibited the expression of FOXD3, which is here characterized as a transcription factor of METTL3, that acts as an m6A writer. By integrating m6A-seq and RNA-seq data after *F. nucleatum* treatment in vitro, the authors identified KIF26B as a potential target gene of METTL3-mediated m6A modification. YTHDF2 was involved in the degradation of methylated KIF26B mRNA. The in vitro and in vivo metastasis models supported that the KIF26B is a potential

oncogene downstream effector of *F. nucleatum* for CRC aggressiveness and metastasis. This is an interesting study. The findings of gut-microbiota on host m6A epitranscriptome and the upstream regulation mechanism of METTL3 by *F. nucleatum* are novel. The experiments are well organized. There are some concerns proposed to improve the manuscript before publication:

1. The m6A modifications of the human cell lines may be affected by the other microbiota, please show that the cells used in this study are mycoplasma free.
2. Is the function mediated by *F. nucleatum* infection dose-dependent?
3. The effects of knocking down KIF26B on metastasis of RKO and SW620 cells in vivo looks dramatic. The author showed a strong reduction in migration of these cells when KIF26B is depleted. I doubt that whether the cells remain healthy when KIF26B is depleted. KIF26B might be an essential gene for these cells, please show the cell viability or proliferation upon knockdown of KIF26B.
4. Recently, some studies reported that the expression levels of METTL3 was increased in CRC and promoted CRC progression, which was contrary to this study, the authors need to explain this controversy in discussion.
5. Please write the full name of 'UHPLC Q-Exact MS analysis' when it first appears.
6. There is a grammar error at the last sentence of Results part. Please delete the 'a'.

Reviewer #3 (Remarks to the Author):

Chen et al. investigated the regulatory mechanisms underlying *F. nucleatum* induced CRC metastasis. They found *F. nucleatum* treatment reduced global m6A modifications in CRC cells and tumor tissues. METTL3 was downregulated upon exposure to *F. nucleatum*, and its m6A methyltransferase activity contributed to *F. nucleatum*-induced CRC aggressiveness. The authors further showed *F. nucleatum* activates YAP signaling meanwhile inhibits FOXD3 expression, and characterized FOXD3 as a transcription factor for METTL3. Downregulation of METTL3 facilitates the expression of target KIF26B, whose expression is critical for *F. nucleatum*-induced CRC metastasis in vivo. The METTL3-FOXD3-KIF26B axis is clinically relevant to CRC. Overall this study elucidated a novel mechanism for *F. nucleatum* induced CRC metastasis. Most data/figures are clear and of high quality and carefully prepared. However, there are some issues that need to be addressed to fully support their conclusions and improve the MS.

1. Figure 1, the authors claimed METTL3 is mediating the *F. nucleatum* -induced metastasis of CRC, it is important to show whether METTL3 knockdown causes the similar phenotype as *F. nucleatum* treatment. Furthermore, *F. nucleatum* induces proliferation as well, the authors completely ignored this issues.
2. Throughout the manuscript, only Transwell assays were used. In many cases invasion capability is also critical for measuring the 'metastasis' potential in vitro. The authors should consider this.
3. Figure 2: D, the YAP nuclear localization staining is not compelling. Better representative images are needed.
4. Figure 2: I, J, *F. nucleatum*-treated conditions should also be included. Also it is surprising that the authors did not measure mRNA levels of METTL3 since they are dealing with transcription of METTL3.
5. This is also true in Fig. 3, there are no mRNA levels of METTL3 upon manipulations of FOXD3/YAP. It is not clear why knockdown of YAP upregulates FOXD3 levels? Do they interact each other? How does YAP inhibit FOXD3?
6. Figure 5, the authors identified KIF26B as a target of METTL3. Since YAP-FOXD3 is controlling METTL3, it is expected to show the KIF26B levels upon YAP/FOXD3 knockdown in these cells.
7. In animal models, why do they implant cells intrasplenically or intravenously, but not orthotopically (into the colon, which is widely used for CRC metastasis)? Do these cells form primary tumors in their systems? If yes, what's the difference between control and knockdown cells? Knockdown of KIF26B is expected to reduce proliferation and tumor formation based on literature. Based on the images (Fig. S6J), knockdown of KIF26B completely blocked tumor formation, irrelevant to metastasis. The same deal in Fig. 6 animal models. These animal models need to be better controlled and explained.
8. Minor, Fig. 7K, the p value and HR number are the same?

REVIEWER COMMENTS

Reviewer #1 (Remarks to the Author):

In this manuscript, Chen and colleagues investigated the role of METTL3 and m6A RNA methylation in colorectal cancer (CRC) metastasis promoted by the pathogenic bacterium *Fusobacterium nucleatum* (*F. nucleatum*). They found that *F. nucleatum* down-regulates the transcription of METTL3, as well as m6A RNA methylation, through activating YAP signaling and subsequently inhibiting FOXD3 activity. Down-regulation of METTL3 increases the mRNA stability of KIF26B, a gene that is critical for CRC metastasis. This manuscript presented novel insights into CRC metastasis under the influence of *F. nucleatum*. However, the main findings of the manuscript is somewhat observational and correlative. Some of the main conclusions are not supported, including (1) the functional significance of METTL3 down-regulation by *F. nucleatum* in CRC aggressiveness, (2) the mechanism by which *F. nucleatum* regulates the YAP signaling, (3) the mechanism by which KIF26B regulates cell migration and CRC metastasis. The following concerns need to be addressed.

Specific comments:

1. One major weakness of the manuscript is the lack of evidence to support the importance of METTL3 down-regulation in CRC aggressiveness, which limits the impact of the manuscript. For example, does knockdown or overexpression of METTL3 affect CRC aggressiveness with or without *F. nucleatum* treatment? Does KIF26B knockdown modulate the effect of METTL3 knockdown/overexpression in CRC aggressiveness with or without the treatment of *F. nucleatum*?

We appreciate the reviewer's suggestions. We have investigated the function of METTL3 in CRC aggressiveness with *F. nucleatum* treatment (**Fig. 1j and Supplementary Fig. 1i**). Here, according to the reviewer's suggestion, we analyzed the effect of overexpressing or knocking down METTL3 on CRC aggressiveness without *F. nucleatum* treatment. Our data showed that overexpression of METTL3 obviously inhibited cell migration of HCT116 cells (**Fig. R1a, b**). Controversely, knockdown of METTL3 enhanced cell migration of HCT116 (**Fig. R1c, d**). These

results suggest that METTL3 functions as a tumor suppressor for CRC aggressiveness. In the current manuscript, we have added **Fig. R1c, d** as **New Supplementary Fig. 1f, g**.

To figure out whether KIF26B mediates the function of METTL3 in CRC aggressiveness, we silenced KIF26B in METTL3-knockdown HCT116 cells with two independent siRNAs (**Fig. R1e**), and found that depletion of KIF26B obviously reversed the enhanced cell migration induced by METTL3 knockdown (**Fig. R1f**). In the current manuscript, we have added **Fig. R1e, f** as **New Fig. 6f, g**.

Fig. R1 METTL3 may function as a CRC tumor suppressor, and KIF26B mediates the function of METTL3 function in CRC aggressiveness.

a, Western blot was performed to detect the expression of METTL3 in HCT116 cells transfected with control or METTL3 plasmid.

b, The HCT116 cells with indicated treatments were applied for transwell migration analysis. Representative images of migrated cells were shown. The migrated cells were quantified in five fields. Scale bar, 100 μ m.

c, Western blot was performed to detect the expression of METTL3 in HCT116 cells transfected with siRNA targeting METTL3 or control siRNAs.

d, HCT116 cells with indicated treatments were subjected to transwell migration analysis. Migrated cells were quantified by counting in five fields. Scale bar, 100 μ m.

e, Western analysis of KIF26B in HCT116 cells transfected with the indicated siRNAs.

f, The HCT116 cells with indicated treatments were applied for transwell migration

analysis. Representative images of migrated cells were shown. The migrated cells were quantified in five fields. Scale bar, 100 μ m.

The western blot results were quantified using Image J software and normalized to the loading control. Data are shown as mean \pm SD. *** $P < 0.001$, **** $P < 0.0001$, by Student's t test.

2. Fig. 1e and extended Fig. 1c: The authors stated that “*F. nucleatum* treatment significantly reduced the expression of METTL3...”. Statistical analysis and SD or SE need to be shown to support this conclusion.

Thanks a lot for the reviewer's valuable concern. According to the reviewer's suggestions, we have repeated the experiments again (Fig. R2a, b) and provided the statistical analysis of METTL3 from three independent experiments (Fig. R2c, d, related to New Supplementary Fig. 1c, e in the current manuscript). In the current manuscript, we have added Fig. R2c, d as New Supplementary Fig. 1c, e.

Fig. R2 *F. nucleatum* treatment significantly reduced the expression of METTL3 in CRC cells.

a, b, Western blot was performed to determine the protein levels of METTL3 in HCT116 cells (a) or LoVo cells (b) treated with *F. nucleatum*, *E. coli* DH5 α or PBS control.

c, d, The statistical analysis of METTL3 protein levels from three independent experiments of HCT116 cells (c) or LoVo cells (d) treated with *F. nucleatum*, *E. coli* DH5 α or PBS control.

The western blot results were quantified using Image J software and normalized to the loading control. Data are shown as mean \pm SD. ns, no significant, ** $P < 0.01$, by Student's t test.

3. Fig. 2a, b: NF- κ B pathway is activated by *F. nucleatum*. Later in extended Data Fig. 2c, the authors showed that knockdown of p65 had not effect on METTL3 expression. However, this is under homeostatis condition, not with the treatment with *F. nucleatum*. They need to test whether NF- κ B inhibition has a role in *F. nucleatum*-induced METTL3 down-regulation.

It is a very good suggestion. Follow the reviewer's instructive suggestions, we treated HCT116 cells with BAY11-7082, which is an inhibitor of NF- κ B, with or without *F. nucleatum* treatment. As the data shown, BAY11-7082 inhibited activation of NF- κ B signaling (**Fig. R3**). However, the BAY11-7082 treatment did not reverse the *F. nucleatum*-induced downregulation of METTL3 (**Fig. R3**). In our current manuscript, we have added **Fig. R3** as **New Supplementary Fig. 2f**.

Fig. R3 NF- κ B inhibition did not reverse the down-regulation of METTL3 induced by *F. nucleatum*. After treating with *F. nucleatum* or PBS control, HCT116 cells were administrated with BAY11-7082 and subjected to western blot analysis for METTL3, p-P65 and P65. The western blot results were quantified using Image J software and normalized to the loading control.

4. Fig. 3e: the effect of FOXD3 on METTL3-luciferase activity is modest.

We thank the reviewer for raising this concern. To evaluate the effect of FOXD3 on the luciferase activity of METTL3, we repeated the dual-luciferase reporter assay again. In the meanwhile, we tested the ability of YAP to affect the luciferase activity of CTGF promoter, which is a well-known target gene of YAP [1], in parallel as a systematic positive control. As the data shown, ectopic expression of FOXD3 significantly increased the luciferase activity of METTL3 promoter (**Fig. R4a**), which is comparable to the effect of YAP on CTGF-luciferase activity (**Fig. R4b**).

Fig. R4 YAP promotes the luciferase activity of *CTGF*-promoter.

a, Dual-luciferase reporter assay showing the effects of FOXD3 overexpression on relative *METTL3*-promoter activity in the HCT116 cells.

b, Dual-luciferase reporter assay showing the effects of YAP overexpression on relative *CTGF*-promoter activity in the HCT116 cells.

Data are shown as mean \pm SD. **** $P < 0.0001$, by Student's t test.

References

[1] Zhao B, *et al.* TEAD mediates YAP-dependent gene induction and growth control. *Genes Dev* **22**, 1962-1971 (2008).

5. Fig. 3I: this figure panel showed that the decreased FOXD3 expression is not important for *F. nucleatum*-induced reduction of *METTL3* expression. What is the mechanism then? How does *F. nucleatum* inhibit the interaction of FOXD3 with the *METTL3* promoter?

We appreciate the reviewer's instructive suggestions. Dual-luciferase reporter assay has suggested that FOXD3 is a positive transcription factor for *METTL3* (**New Fig. 3g in the current manuscript**). To make this conclusion more convincing, we used qRT-PCR to detect the mRNA levels of *METTL3* in FOXD3-overexpression HCT116 cells (**Fig. R5a, related to New Supplementary Fig. 3a in the current manuscript**) and FOXD3-knockdown HCT116 cells (**Fig. R5b, related to New Supplementary Fig. 3b in the current manuscript**). These results supported our previous conclusion.

The data we shown in **New Fig. 3i**, is a Dual-luciferase reporter assay to test the ability of FOXD3 to promote *METTL3* transcription with or without *F. nucleatum* treatment. Under these two different conditions, we found that the transcriptional

promotion of *METTL3* by *FOXD3* significantly reduced in the presence of *F. nucleatum*. And the result of ChIP assay (New Fig. 3h in the current manuscript) also suggested that *F. nucleatum* treatment could influence the interaction between *FOXD3* and *METTL3* promoter. Mechanistically, our results showed that *F. nucleatum* can downregulate the expression of *FOXD3* (New Fig. 3j, k and New Supplementary Fig. 3c in the current manuscript) through YAP signaling (New Fig. 3l, m). Here, we further predicted the potential binding sites of YAP-TEAD on the promoter of *FOXD3* and the ChIP assay confirmed the binding potential (Fig R5c, d). Dual-luciferase reporter assay confirmed that the knockdown of YAP significantly enhanced *FOXD3*-luciferase activity in HCT116 cells (Fig R5e). Thus, we propose that *F. nucleatum* downregulates *METTL3* may through YAP-mediated inhibition of *FOXD3*.

Fig. R5 FOXD3 regulates the mRNA level of *METTL3*.

a, Quantitative RT-PCR was performed in HCT116 cells to detect the expression of *METTL3* mRNA after transfection with *FOXD3* plasmid or control vector.

b, Quantitative RT-PCR was performed in HCT116 cells to detect the expression of *METTL3* mRNA after transfection with siRNA targeting *FOXD3*.

c, Schematic illustration of the potential binding sites of YAP-TEAD on the promoter of *FOXD3*. Five pairs of primers were showed for following ChIP analysis.

d, ChIP-qPCR analysis of YAP-TEAD binding to the predicted binding regions of *FOXD3* promoter in HCT116 cells. The ChIP-qPCR results were presented as

enrichment of YAP-TEAD at FOXD3 promoter relative to IgG. Positive control (CYR61 and CTGF).

e, Dual-luciferase reporter assay showing the effects of YAP knockdown on relative *FOXD3*-promoter (-2000 bp-100 bp) activity in the HCT116 cells.

Data are shown as mean \pm SD. *** $P < 0.001$, **** $P < 0.0001$, by Student's t test. N/A, not applicable.

6. Fig. 5c: what are the peaks before the stop codon?

We are grateful to reviewer for pointing out this concern. We consulted the technicians of the company, where we performed the m⁶A-Seq in this study. They confirmed that the peaks before the stop codon were background noise generated during sequencing.

7. Fig. 5F: What criteria is used to justify “strong interaction between KIF26B mRNA and METTL3.”.

We thank the reviewer for raising this concern. In fact, when we performed this assay, METTL3 target gene *CREBBP* was served as a positive control while *HPRT1* was served as a negative control [1]. The interaction between METTL3 and *KIF26B* mRNA is comparable to the interaction between METTL3 and its well-known target *CREBBP* mRNA (Fig. R7). In our current manuscript, we have added Fig. R7 as the New Fig. 5f.

Fig. R7 RNA Immunoprecipitation (RIP)-qPCR analysis of METTL3 binding with *KIF26B* mRNA, or *CREBBP* mRNA (positive control) or *HPRT1* mRNA (negative control) in HCT116 cells.

Data are shown as mean \pm SD. ** $P < 0.01$, **** $P < 0.0001$, by Student's t test.

References

[1] Wang X, *et al.* N6-methyladenosine-dependent regulation of messenger RNA stability. *Nature* **505**, 117-120 (2014).

8. Fig. 5n, o: does YTHDF1 knockdown affect the mRNAs stability of KIF26B with or without *F. nucleatum* treatment? Knockdown of YTHDF1-3 seems to have similar effect on KIF26B levels to YTHDF2 knockdown. How about knockdown of each YTHDF protein? Does knockdown of YTHDF1 or YTHDF3 affect the mRNA stability of KIF26B?

We thank the reviewer for raising this concern. According to reviewer's suggestion, we performed RNA stability assay to analyze whether knockdown YTHDF1 or YTHDF3 affects the mRNA stability of KIF26B in CRC cells with or without *F. nucleatum* treatment. Our results suggested the silence of YTHDF1 or YTHDF3 had no significant effects on *KIF26B* mRNA stability (**Fig. R8a-d**). In addition, we also observed that depletion of YTHDF1 (**Fig. R8e**) or YTHDF3 (**Fig. R8f**) had no significant effects on the expression levels of *KIF26B* mRNA. In our current manuscript, we have added **Fig. R8e, f** as the **New Supplementary Fig. 5h, i**.

Fig. R8 The downregulation of YTHDF1 or YTHDF3 shown no effect on *KIF26B* mRNA stability neither with *F. nucleatum* treatment nor without.

a, b, HCT116 cells were transfected with siRNA targeting YTHDF1(**a**) or YTHDF3 (**b**) for 48 h. The remaining levels of *KIF26B* mRNAs were analyzed by quantitative RT-PCR at the indicated time points after actinomycin D treatment.

c, d, HCT116 cells were transfected with siRNA targeting YTHDF1(**c**) or YTHDF3 (**d**) for 36 h, then treated with *F. nucleatum* for 12 h. The remaining levels of *KIF26B* mRNAs were analyzed by quantitative RT-PCR at the indicated time points after actinomycin D treatment.

e, f, Quantitative RT-PCR was performed in HCT116 cells to detect the expression of *KIF26B* mRNA after transfection with siRNA targeting YTHDF1 (**e**) or YTHDF3 (**f**). Data are shown as mean \pm SD. ns, no significant, **** $P < 0.0001$, by Student's t test.

9. How does *F. nucleatum* activate YAP signaling?

This is a very interesting question. In this study, we demonstrated that *F. nucleatum* could activate YAP signaling. We also observed that *F. nucleatum* treatment reduced both MST1/2 phosphorylation and LATS1/2 phosphorylation levels (**New Fig. 2 in the current manuscript**), which are the major members of Hippo cascade.

Here, according to the reviewer's suggestion, we further tried to analyze the upstream events of Hippo pathway. NF2, KIBRA, and WILLIN\FRMD6 are well-known classical activators for MST1/2 [1-2]. We detected the levels of NF2, KIBRA and FRMD6 in *F. nucleatum*-treated HCT116 cells or LoVo cells. Interestingly, *F. nucleatum* treatment notably inhibited the levels of these three activators in both two CRC cells (**Fig. R9a, b**). Collectively, these results suggest that *F. nucleatum* treatment activates YAP signaling through inhibition of the Hippo pathway in CRC cells. It will be of interest to further study the detail mechanism for gut microbes modulating host Hippo signaling. In our current manuscript, we have added **Fig. R9a, b** as the **New Supplementary Fig. 2b, c**.

Fig. R9 *F. nucleatum* downregulated the expression of MST activators of Hippo-YAP pathway.

a, b, Western blot was performed to detect the expression levels of NF2, KIBRA and FRMD6 in HCT116 cells (**a**) or LoVo cells (**b**) treated with *F. nucleatum* or PBS control. The western blot results were quantified using Image J software and normalized to the loading control.

References

[1] Grusche FA, Richardson HE, Harvey KF. Upstream regulation of the hippo size control pathway. *Curr Biol* **20**, R574-582 (2010).

[2] Moya IM, Halder G. Hippo-YAP/TAZ signalling in organ regeneration and regenerative medicine. *Nat Rev Mol Cell Biol* **20**, 211-226 (2019).

10. Fig. 2h, 2i, 3b, 3c, 3f, 3i: the m⁶A levels need to be determined when the upstream signaling for METTL3 expression is manipulated to support the importance of these METTL3 regulators in controlling m⁶A enrichment.

This is very a good suggestion. According to reviewer's instructive suggestions, we overexpressed or knocked down the expression of YAP or FOXD3 in CRC cells. Dot-blot assay showed that the m⁶A levels were increased when YAP was silenced (**Fig. R10a**). Conversely, the overexpression of YAP declined the m⁶A levels (**Fig. R10b**). These results were consistent with the effects of YAP on METTL3 levels (**Fig. R10c, d, related to New Fig. 2h, j in the current manuscript**), suggesting that YAP signaling is an upstream regulator for METTL3 to control m⁶A enrichment in CRC cells. Similarly, depletion of FOXD3 decreased m⁶A levels (**Fig. R10e**) while overexpression of FOXD3 increased the m⁶A levels (**Fig. R10f**), resembling the effects of FOXD3 on METTL3 levels (**Fig. R10g, h, related to New Fig. 3c, b in the current manuscript**). Taken together, these results suggest that YAP and FOXD3 act as upstream regulators of METTL3 in controlling m⁶A enrichment. In the current manuscript, we have added **Fig. 10a, b** as the **New Fig. 2i, k**, added **Fig. 10e, f** as the **New Fig. 3e, d**.

Fig. R10 YAP and FOXD3 act as upstream regulators of METTL3 in controlling m⁶A enrichment.

a, mRNA dot blot analysis was performed to determine the m⁶A levels of HCT116 cells after transfection with siRNAs targeting YAP.

b, mRNA dot blot analysis was performed to determine the m⁶A levels of HCT116 cells after transfection with YAP plasmid or control vector.

c, HCT116 cells with the indicated treatment were subjected to western blot analysis of METTL3 or GAPDH.

d, HCT116 cells transfected with YAP overexpression or control pcDNA3.1 plasmids were subjected to western blot analysis of METTL3 or GAPDH.

e, mRNA dot blot analysis was performed to determine the m⁶A levels of HCT116 cells after transfection with shRNAs targeting FOXD3.

f, mRNA dot blot analysis was performed to determine the m⁶A levels of HCT116 cells after transfection with FOXD3 plasmid or control vector.

g, Western blot analysis of METTL3 in HCT116 cells transfected with two different shRNAs targeting FOXD3 or control shRNAs in pLKO.1 vector.

h, Western blot analysis of METTL3 in HCT116 cells transfected with FOXD3-myc-his vector or pcDNA3.1(+)/myc-His C control vector.

The methylene blue staining was used as a loading control. The western blot results and the dot blot results were quantified using Image J software and normalized to the loading control.

Reviewer #2 (Remarks to the Author):

In this study, the authors found that colorectal cancer enriched *Fusobacterium nucleatum*, a well-known oncogenic bacterium, suppressed m⁶A levels in CRC cell lines and CRC patient cohort. It was further indicated that *F. nucleatum*, through the inhibition of the Hippo pathway, causes activation of YAP. YAP activation inhibited the expression of FOXD3, which is here characterized as a transcription factor of METTL3, that acts as an m⁶A writer. By integrating m⁶A-seq and RNA-seq data after *F. nucleatum* treatment in vitro, the authors identified KIF26B as a potential target gene of METTL3-mediated m⁶A modification. YTHDF2 was involved in the degradation of methylated KIF26B mRNA. The in vitro and in vivo metastasis models supported that

the KIF26B is a potential oncogene downstream effector of *F. nucleatum* for CRC aggressiveness and metastasis. This is an interesting study. The findings of gut-microbiota on host m6A epitranscriptome and the upstream regulation mechanism of METTL3 by *F. nucleatum* are novel. The experiments are well organized. There are some concerns proposed to improve the manuscript before publication:

1. The m6A modifications of the human cell lines may be affected by the other microbiota, please show that the cells used in this study are mycoplasma free.

We thank the reviewer for raising this concern. According to reviewer's instructive suggestions, we have detected whether there existed mycoplasma in our CRC cell lines, as the data shown, the CRC cell lines we used in our study were all mycoplasma free (**Fig. R11**).

Mycoplasma detection

Fig. R11 The CRC cell lines used in this study were mycoplasma free.

RT-PCR analysis for the mycoplasma in the medium of HCT116, LoVo, RKO and SW620 cell lines. Positive control, culture medium with mycoplasma; Negative control, sterile ddH₂O.

2. Is the function mediated by *F. nucleatum* infection dose-dependent?

We appreciate reviewer's instructive suggestions. To figure out whether the improvement of the migration induced by *F. nucleatum* is dose-dependent, different multiplicity of infection (MOI) of *F. nucleatum* was used in the migration assay. As the data shown, when we treated CRC cells with *F. nucleatum* at the MOI of 50:1, 100:1 and 200:1, *F. nucleatum* can significantly increase the migration ability of CRC cell. But we did not observe the dose-dependent manner (**Fig. R12**).

Fig. R12 Migration assay was performed in LoVo cells treated with different MOI of *F. nucleatum*. LoVo cells were pretreated with indicated MOI of *F. nucleatum* or PBS control for 2 h and subjected to transwell assay (Left). The migrated cells were quantified by counting in five fields (Right). Scale bar, 100 μ m. Data are shown as mean \pm SD. ** $P < 0.01$, by Student's t test.

3. The effects of knocking down KIF26B on metastasis of RKO and SW620 cells in vivo looks dramatic. The author showed a strong reduction in migration of these cells when KIF26B is depleted. I doubt that whether the cells remain healthy when KIF26B is depleted. KIF26B might be an essential gene for these cells, please show the cell viability or proliferation upon knockdown of KIF26B.

We thank the review for raising this important concern. In this study, we revealed the essential role of KIF26B in CRC metastasis. In fact, the CRC cells remained healthy when we transfected cells with KIF26B siRNA or shRNAs. According to the reviewer's suggestion, we tested whether KIF26B knockdown affected cell proliferation in the four CRC cell lines we used in our manuscript. Our results showed that the depletion of KIF26B has mild effect on the proliferation of CRC cells *in vitro* (**Fig. R13a-d**). Moreover, we established stable KIF26B-knockdown HCT116 and RKO cell lines (**Fig. R13e, f**) and these cell lines were subcutaneously injected into nude mice. As the data shown in **Fig. R13g-i**, we did not observe a dramatic inhibition of tumor growth caused by depletion of KIF26B. Similar result was also observed in RKO cells that downregulation of KIF26B has weak effect on tumor formation (**Fig. 13j-l**). Taken together, these results indicate that the dramatic inhibition of CRC metastasis caused by knockdown of KIF26B is not due to its effect on CRC cell proliferation and tumor growth.

Fig. R13 The downregulation of KIF26B have weak effect on cell proliferation and tumor formation in CRC.

a-d, MTT analysis of HCT116 cells (**a**), LoVo cells (**b**), RKO cells(**c**) or SW620 cells (**d**) transfected with control or KIF26B siRNA.

e, f, Western blot analysis of KIF26B expression in HCT116 cells (**e**) or RKO cells (**f**) infected with two different shRNAs targeting KIF26B or control shRNA.

g, Control and KIF26B-silenced HCT116 cells were subcutaneously injected into BALB/C nude mice. The images of tumors in each group are presented. Scale bar, 1 cm.

h, The tumor volume was measured at indicated time, and the growth curves are shown.

i, The tumor weight was measured after the mice were sacrificed.

j, Control and KIF26B-knockdown RKO cells were subcutaneously injected into BALB/C nude mice. The images of tumors in each group are presented. Scale bar, 1 cm.

k, The tumor volume was measured at indicated time, and the growth curves are shown.

l, The tumor weight was measured after the mice were sacrificed.

Data are shown as mean \pm SD. ns, no significant, * $P < 0.05$, *** $P < 0.001$, by Mann-Whitney test.

4. Recently, some studies reported that the expression levels of METTL3 was increased in CRC and promoted CRC progression, which was contrary to this study, the authors need to explain this controversy in discussion.

We thank the review for raising this concern. Indeed, we also noticed that there are contradictory reports about the effects of METTL3 on CRC progression [1]. Jun Yu group also found that METTL3 enhanced CRC tumorigenesis and tumor growth by activating m⁶A-GLUT1-mTORC1 axis [2]. However, Yan-hong Deng group observed the tumor suppressor function of METTL3 in migration and invasion of CRC cells through blocking p38/ERK pathway [3]. It seems that METTL3 promotes CRC tumor growth while suppresses CRC cell migration and metastasis. Since METTL3 is the most important methyltransferase for mRNA m⁶A modifications which play diverse functions in many biology process and disease, it is possible that METTL3 acts either oncogenic or tumor suppressive roles in different stages of CRC progression by modulating different target pathways. Here, we revealed that the gut microbes *F. nucleatum* can reduce the m⁶A levels of CRC cells through inhibition of METTL3, leading to enhanced CRC cell migration and metastasis. Under the condition of *F. nucleatum* treatment, METTL3 acted as a tumor suppressor gene for CRC metastasis by downregulation of target KIF26B expression. However, the diverse functions and mechanisms of METTL3 in CRC progression as well as other cancers need further studies. According to the reviewer's suggestion, we have explained this point in the discussion section in the current manuscript.

References

[1] Li T, *et al.* METTL3 facilitates tumor progression via an m(6)A-IGF2BP2-dependent mechanism in colorectal carcinoma. *Mol Cancer* **18**, 112 (2019).

[2] Chen H, *et al.* RNA m6A methyltransferase METTL3 facilitates colorectal cancer by activating m6A-GLUT1-mTORC1 axis and is a therapeutic target. *Gastroenterology*, (2020).

[3] Deng R, *et al.* m(6)A methyltransferase METTL3 suppresses colorectal cancer proliferation and migration through p38/ERK pathways. *Onco Targets Ther* **12**, 4391-4402 (2019).

5. Please write the full name of 'UHPLC Q-Exactive MS analysis' when it first appears. We apologize for not making this clear in the manuscript and thank the reviewer for raising this point. The full name of 'UHPLC Q-Exactive MS' is 'ultra – high - performance liquid chromatography coupled with quadrupole - Exactive mass spectrometry', and this information has been added in our current manuscript.

6. There is a grammar error at the last sentence of Results part. Please delete the 'a'. We apologize for such grammar error and have corrected this error in the current manuscript. We thank the reviewers for helping us improve our manuscript.

Reviewer #3 (Remarks to the Author):

Chen et al. investigated the regulatory mechanisms underlying F. nucleatum induced CRC metastasis. They found F. nucleatum treatment reduced global m6A modifications in CRC cells and tumor tissues. METTL3 was downregulated upon exposure to F. nucleatum, and its m6A methyltransferase activity contributed to F. nucleatum-induced CRC aggressiveness. The authors further showed F. nucleatum activates YAP signaling meanwhile inhibits FOXD3 expression, and characterized FOXD3 as a transcription factor for METTL3. Downregulation of METTL3 facilitates the expression of target KIF26B, whose expression is critical for F. nucleatum-induced CRC metastasis in vivo. The METTL3-FOXD3-KIF26B axis is clinically relevant to CRC.

Overall this study elucidated a novel mechanism for F. nucleatum induced CRC metastasis. Most data/figures are clear and of high quality and carefully prepared.

However, there are some issues that need to be addressed to fully support their conclusions and improve the MS.

1. Figure 1, the authors claimed METTL3 is mediating the *F. nucleatum* –induced metastasis of CRC, it is important to show whether METTL3 knockdown causes the similar phenotype as *F. nucleatum* treatment. Furthermore, *F. nucleatum* induces proliferation as well, the authors completely ignored this issue.

We appreciate the reviewer’s suggestions. According to the reviewer’s suggestion, we tested the effect of METTL3 knockdown on the cell migration of CRC cells without *F. nucleatum* treatment (**Fig. R14a**). Our data showed that depletion of METTL3 obviously promoted cell migration of HCT116 cells (**Fig. R14b**), which is similar with the phenotype caused by *F. nucleatum* treatment. In our current manuscript, we have added **Fig. R14a, b** as the **New Supplementary Fig. 1f, g**.

Regarding the role of *F. nucleatum* in CRC cell proliferation, it has been many studies reported that *F. nucleatum* treatment promoted CRC cell proliferation and tumor growth [1, 2]. Here, according to the reviewer’s suggestion, we tested the cell proliferation and also observed the similar phenomena that *F. nucleatum*-treatment indeed could promote cell proliferation of CRC cells (**Fig. R14c**). It will be interesting to explore whether METTL3 is involved in *F. nucleatum*-induced cell proliferation and its underlying mechanism. We have added this point in the discussion section of this current manuscript.

Fig. R14 Knockdown of METTL3 promotes CRC cells migration and *F. nucleatum*-treatment enhance cell proliferation.

a, Western blot was performed in HCT116 cells to detect the expression of METTL3 after transfected with siRNA targeting METTL3 or control siRNAs.

b, HCT116 cells were transfected with siRNA targeting METTL3 or control siRNAs and subjected to transwell migration analysis. Migrated cells were quantified by counting in five fields. Scale bar, 100 μ m.

c, HCT116 cells were incubated with PBS or *F. nucleatum* for 2 h. The cell proliferation rates were evaluated by CCK8 assay at 4, 24, 48, 72 and 96 h.

Data are shown as mean \pm SD. ns, no significant, ** $P < 0.01$, *** $P < 0.001$, **** $P < 0.0001$, by Student's t test.

References

[1] Rubinstein MR, Wang X, Liu W, Hao Y, Cai G, Han YW. Fusobacterium nucleatum promotes colorectal carcinogenesis by modulating E-cadherin/beta-catenin signaling via its FadA adhesin. *Cell Host Microbe* **14**, 195-206 (2013).

[2] Yang Y, *et al.* Fusobacterium nucleatum Increases Proliferation of Colorectal Cancer Cells and Tumor Development in Mice by Activating Toll-Like Receptor 4 Signaling to Nuclear Factor-kappaB, and Up-regulating Expression of MicroRNA-21. *Gastroenterology* **152**, 851-866 e824 (2017).

2. Throughout the manuscript, only Transwell assays were used. In many cases invasion capability is also critical for measuring the 'metastasis' potential in vitro. The authors should consider this.

Thanks for the reviewer's valuable advice. According to the reviewer's suggestion, we performed invasion assay to further evaluate the metastasis. The results showed that *F. nucleatum*, but not *E. coli* significantly promoted cell invasion of both HCT116 cells (**Fig. R15a**) and LoVo cells (**Fig. R15b**). Similarly, knockdown of KIF26B dramatically inhibited the cell invasion of HCT116 cells (**Fig. R15c**) and LoVo cells (**Fig. R15d**).

Fig. R15 *F. nucleatum* treatment promotes CRC cells invasion, and the downregulation of KIF26B inhibits CRC cells invasion.

a, b, HCT116 cells (**a**) or LoVo cells (**b**) were pretreated with *F. nucleatum*, *E. coli* DH5 α or PBS control for 2 h and subjected to invasion assay (Left). The invaded cells were quantified by counting in five fields (Right). Scale bar, 100 μ m.

c, d, HCT116 cells (**c**) or LoVo cells (**d**) were transfected with two siRNAs targeting KIF26B or control siRNAs and subjected to invasion analysis (Left). The invaded cells were quantified by counting in five fields (Right). Scale bar, 100 μ m.

Data are shown as mean \pm SD. *** $P < 0.001$, **** $P < 0.0001$, by Student's t test.

3. Figure 2: D, the YAP nuclear localization staining is not compelling. Better representative images are needed.

Thanks for helping us improve our manuscript. According to the reviewer's advice, we repeated this experiment again and provided clearer representative images (**Fig. R16**).

We have added **Fig. R16** as the New **Fig. 2d**.

Fig. R16 *F. nucleatum* treatment promotes YAP nuclear translocation.

a, Immunofluorescence analysis of the YAP distribution in the indicated cells. Cells were stained with specific antibody against YAP (red), and nuclei were counterstained with DAPI (blue). Scale bar, 20 μ m.

b, The percentage of nuclear YAP were quantified by counting in ten fields.

Data are shown as mean \pm SD. **** $P < 0.0001$, by Student's t test.

4. Figure 2: I, J, *F. nucleatum*-treated conditions should also be included. Also it is surprising that the authors did not measure mRNA levels of METTL3 since they are dealing with transcription of METTL3.

We really appreciate the reviewer's wise advices. Here, we verified the role of YAP in regulating the expression of METTL3 in the presence of *F. nucleatum*. Consistent with the previous results, *F. nucleatum*-treatment significantly decreased the expression of METTL3, while YAP here as an upstream regulator of METTL3, the depletion of YAP almost entirely rescued the down-regulation of METTL3 expression induced by *F. nucleatum* (**Fig. R17**). Collectively, our data further indicate that YAP signaling is involved in *F. nucleatum*-induced downregulation of METTL3 expression in CRC cells. We have added **Fig. R17** as the **New Fig. 2m** in the current manuscript.

Fig. R17 YAP mediate the downregulation of METTL3 in *F. nucleatum*-treated CRC cells. Western blot analysis of METTL3 in HCT116 cells transfected with the indicated siRNAs. The western blot results were quantified using Image J software and normalized to the loading control.

As for the detection of *METTL3* mRNA levels, we thank the reviewer for helping us improve our manuscript. Here, we overexpressed or knocked down the expression of YAP or FOXD3, which are the upstream regulators of *METTL3*, then the mRNA levels of *METTL3* were analyzed. *METTL3* mRNA levels were significantly decreased when YAP was overexpressed (**Fig. R18a**). Conversely, the silence of YAP upregulated *METTL3* mRNA levels, suggesting that YAP signaling is a negative upstream regulator for *METTL3* in CRC cells (**Fig. R18b**). We also observed that FOXD3 increased *METTL3* mRNA levels while depletion of FOXD3 decreased *METTL3* mRNA levels (**Fig. R18c, d**). We have added **Fig. R18 b** as **New Fig. 3I**, added **Fig. R18c, d** as **New Supplementary Fig. 3a, b** in the current manuscript.

Fig. R18 YAP and FOXD3 as the upstream regulators of METTL3 participate in its transcriptional regulation.

a, Quantitative RT-PCR was performed in HCT116 cells to detect the expression of *FOXD3* mRNA and *METTL3* mRNA after transfection with YAP plasmid or control

vector.

b, Quantitative RT-PCR was performed in HCT116 cells to detect the expression of *FOXD3* mRNA and *METTL3* mRNA after transfection with siRNA targeting YAP.

c, Quantitative RT-PCR was performed in HCT116 cells to detect the expression of *METTL3* mRNA after transfection with *FOXD3* plasmid or control vector.

d, Quantitative RT-PCR was performed in HCT116 cells to detect the expression of *METTL3* mRNA after transfection with siRNA targeting *FOXD3*.

Data are shown as mean \pm SD. ** $P < 0.01$, *** $P < 0.001$, **** $P < 0.0001$, by Student's t test.

5. This is also true in Fig. 3, there are no mRNA levels of *METTL3* upon manipulations of *FOXD3*/*YAP*. It is not clear why knockdown of *YAP* upregulates *FOXD3* levels? Do they interact each other? How does *YAP* inhibit *FOXD3*?

We thank the reviewer for raising these important concerns. According to the reviewer's suggestion, we overexpressed or knocked down the expression of *YAP* or *FOXD3*, then the mRNA levels of *METTL3* were analyzed. *METTL3* mRNA levels were significantly decreased when *YAP* was overexpressed (**Fig. R19a**). Conversely, knockdown of *YAP* upregulated *METTL3* mRNA levels (**Fig. R19b**). Ectopic expression of *FOXD3* increased *METTL3* mRNA levels while depletion of *FOXD3* decreased *METTL3* mRNA levels (**Fig. R19c, d**). We have added **Fig. R19b** as **New Fig. 3I**, added **Fig. R19c, d** as **New Supplementary Fig. 3a, b** in the current manuscript.

For the relationship between *YAP* and *FOXD3*, our previous western blot analysis showed that knockdown of *YAP* upregulates *FOXD3* levels. Here, we observed that *YAP* regulated the mRNA levels of *FOXD3* (**Fig. R19a, b**). Moreover, we used the online bioinformatics tools *JASPAR* to analyze the relationship between *YAP* signaling and *FOXD3*. The prediction results showed that there exist multiple binding sites of *YAP*-TEAD in the promoter region (-2000 bp ~ 200 bp) of *FOXD3* (**Fig. R19e**). Chromatin immunoprecipitation (ChIP) assays were employed (**Fig. R19f**) to verify whether *YAP*-TEAD is an upstream regulator for *FOXD3* transcription using *YAP*-antibody according to previous report [1]. The ChIP assays revealed that *YAP* can bind to the promoter region of *FOXD3*. The regions of -1587 bp ~ -1425 bp containing two binding sites (*YAP*-TEAD-Primer1), -1098 bp ~ -965 bp containing one binding site (*YAP*-TEAD-Primer2) and -592 bp ~ -486 bp containing two binding sites (*YAP*-

TEAD-Primer3), may be the potential binding sites for YAP-TEAD co-transcription factor, while the regions of -418 bp ~ -268 bp containing two binding sites (YAP-TEAD-Primer4) and -11 bp ~ 106 bp containing two binding sites (YAP-TEAD-Primer5) were not detectable (**Fig. R19g**). The region of -592 bp ~ -486 bp showed an obvious binding potential comparable to the other two well-known positive control (CYR61 and CTGF) [1]. Furthermore, a dual-luciferase reporter assay confirmed that the knockdown of YAP significantly enhanced FOXD3-luciferase activity in HCT116 cells (**Fig. R19h**). Collectively, our results suggest that YAP signaling can inhibit the expression of FOXD3.

Regarding whether YAP can bind with FOXD3, the nucleus fractions were isolated from CRC cells, then we performed co-immunoprecipitation assay. We observed a mild interaction between FOXD3 and YAP using either FOXD3 antibodies or YAP antibodies (**Fig. R19i**). We noticed that recent studies have reported that YAP can bind with some transcription factors to influence their transcriptional regulatory function [2, 3]. For instance, Yong-yu Wang group observed that the activated YAP can interacted with STAT3 in the nucleus, which is a transcription factor of VEGF, to inhibit VEGF expression [3]. It is possible that the *F. nucleatum*-induced activated YAP activation in the nucleus may also bind to FOXD3, thereby inhibiting the transcription METTL3.

Fig. R19 YAP and FOXD3 as the upstream regulators of METTL3 participate in its transcriptional regulation, and nuclear YAP inhibits the transcription of FOXD3.

a, Quantitative RT-PCR was performed in HCT116 cells to detect the expression of *FOXD3* mRNA and *METTL3* mRNA after transfection with YAP plasmid or control vector.

b, Quantitative RT-PCR was performed in HCT116 cells to detect the expression of *FOXD3* mRNA and *METTL3* mRNA after transfection with siRNA targeting YAP.

c, Quantitative RT-PCR was performed in HCT116 cells to detect the expression of

METTL3 mRNA after transfection with FOXD3 plasmid or control vector.

d, Quantitative RT-PCR was performed in HCT116 cells to detect the expression of *METTL3* mRNA after transfection with siRNA targeting FOXD3.

e, The prediction results of TEAD binding site on FOXD3 promoter (top 18) using the online bioinformatics tools *JASPAR*.

f, Schematic illustration of the potential binding sites of YAP-TEAD on the promoter of FOXD3. Five pairs of primers were showed for following ChIP analysis.

g, ChIP-qPCR analysis of YAP-TEAD binding to the predicted binding regions of FOXD3 promoter in HCT116 cells. The ChIP-qPCR results were presented as enrichment of YAP-TEAD at FOXD3 promoter relative to IgG.

h, Dual-luciferase reporter assay showing the effects of YAP knockdown on relative *FOXD3*-promoter (-2000 bp-100 bp) activity in the HCT116 cells.

i, The nucleus fractions were isolated from HCT116 cells, and YAP-antibody (Left) or FOXD3-antibody (Right) were used for co-immunoprecipitation assay, respectively. Western blot analysis of the interaction between YAP and FOXD3 in the nucleus.

Data are shown as mean \pm SD. ** $P < 0.01$, *** $P < 0.001$, **** $P < 0.0001$, by Student's t test. N/A, not applicable.

References

[1] Kim T, *et al.* A basal-like breast cancer-specific role for SRF-IL6 in YAP-induced cancer stemness. *Nat Commun* **6**, 10186 (2015).

[2] Wang L, *et al.* Yes-Associated Protein Inhibits Transcription of Myocardin and Attenuates Differentiation of Vascular Smooth Muscle Cell from Cardiovascular Progenitor Cell Lineage. *Stem Cells* **35**, 351-361 (2017).

[3] Fan X, *et al.* YAP promotes endothelial barrier repair by repressing STAT3/VEGF signaling. *Life Sci* **256**, 117884 (2020).

6. Figure 5, the authors identified KIF26B as a target of METTL3. Since YAP-FOXD3 is controlling METTL3, it is expected to show the KIF26B levels upon YAP/FOXD3 knockdown in these cells.

We really appreciate the reviewer's suggestion. Here, we depleted YAP with two independent siRNAs. KIF26B protein levels were significantly decreased when YAP was silenced (**Fig. R20a**). In addition, the expression of FOXD3 was silenced by two

shRNAs, then the levels of KIF26B were also detected by western blot. As the data showed in **Fig. R20b**, the knockdown of FOXD3 significantly induced KIF26B expressing. We have added **Fig. R20** as the **New Supplementary Fig. 5e, f** in the current manuscript.

Fig. R20 YAP and FOXD3 could mediate the expression of KIF26B.

a, HCT116 cells with the indicated treatment were subjected to western blot analysis of KIF26B.

b, HCT116 cells transfected with shRNAs targeting FOXD3 or control shRNA were subjected to western blot analysis of KIF26B.

The western blot results were quantified using Image J software and normalized to the loading control.

7. In animal models, why do they implant cells intrasplenically or intravenously, but not orthotopically (into the colon, which is widely used for CRC metastasis)? Do these cells form primary tumors in their systems? If yes, what's the difference between control and knockdown cells? Knockdown of KIF26B is expected to reduce proliferation and tumor formation based on literature. Based on the images (Fig. S6J), knockdown of KIF26B completely blocked tumor formation, irrelevant to metastasis. The same deal in Fig. 6 animal models. These animal models need to be better controlled and explained.

We thank the reviewer for raising this concern. In fact, the intravenous injection model and intrasplenic injection model were also commonly used in studying CRC metastasis [1-5]. However, we fully agree with the reviewer that orthotopic inoculation model would be the best. According to reviewer's instructive suggestions, we followed the protocol from previous studies [6-7], and tried our best to establish an orthotopic model to validate our *in vivo* results.

First, we established stable KIF26B-knockdown luciferase-labeled HCT116 cells (**Fig. R21a**). The control HCT116-luc or KIF26B-knockdown HCT116-luc were incubated with *E. coli* or *F. nucleatum* (MOI of 100:1) for 24 h, and orthotopically injected into NOD SCID mice (2×10^6 cells in 50 μ l /per mouse, PBS: Matrigel =1:1). Next day, we used live imaging to confirm whether modeling successfully (**Fig. R21c, above**). It is a pity that many mice suffered intestinal obstruction and died in succession within one month. We have to terminate the assay at the day of 45 after modeling. The mice were sacrificed. We dissected the colon, as well as peritoneal tumor tissues, of the mice (**Fig. R21c, below**) and found that the knockdown of KIF26B did not dramatically affect primary tumor growth. We did not observe macroscopic metastases in the liver of each mice, which probably due to the modeling time. So, we performed H&E staining on the whole liver of each mice to assess the micro-metastasis. Interestingly, the mice receiving *F. nucleatum*-treated cells obviously developed more micro-metastases than those receiving untreated or *E. coli*-treated control cells (**Fig. R21d, e**). Knockdown of KIF26B notably diminished *F. nucleatum*-induced liver metastasis (**Fig. R21d, e**), which was similar with the results of the intrasplenic injection model in **New Fig. 6j, k**. We have added **Fig. R21a, d and e** as **new Supplementary Fig. 7 c-e** in the current manuscript.

As for whether the knockdown of KIF26B will affect cell proliferation, we thank the review for raising this important concern. Besides the orthotopic injection model, we subcutaneously injected control HCT116-luc cell line and two KIF26B-knockdown HCT116 cells into nude mice to prove whether KIF26B knockdown affects tumor formation. As the data shown in **Fig. R21f-h**, we did not observe the dramatic inhibition of tumor growth upon depletion of KIF26B. Similar with the result of HCT116, the downregulation of KIF26B has no significant effect on the tumor growth of RKO cells (**Fig. R21b, R21i-k**). Collectively, although KIF26B was reported to affect cell proliferation in breast cancer [8] or gastric cancer [9], we did not observe similar phenomena in cell proliferation and tumor growth of CRC cells. The dramatic decrease of CRC cell migration and metastasis caused by KIF26B knockdown is not due to its inhibition in CRC cell proliferation.

Regarding the mentioned results shown in **Fig. S6J (related to New Supplementary Fig. 7a, b in the current manuscript)** is actually an intravenous injection model, but not a subcutaneous model. The images of control group showed

the formation of bone metastasis in mice. Sorry we did not present data clearly. We have added the information in the figure legends of the current manuscript.

Fig. R21 *F. nucleatum* accelerates CRC aggressiveness and metastasis by upregulating KIF26B and the downregulation of KIF26B have little effect on tumor formation.

a, b, Western blot analysis of KIF26B expression in HCT116-luc cells (**a**) or RKO cells (**b**) infected with two different shRNAs targeting KIF26B or control shRNA.

c, Luciferase-labeled HCT116 cells were stably infected with lentivirus-based KIF26B shRNAs or control shRNAs. The indicated cells were co-cultured with *E. coli* or *F. nucleatum* for 24 h and orthotopically injected into Nod Scid mice. Representative bioluminescence image of orthotopical injection mice model (Above). The gross colons of each mice in five indicated groups (Below). Scale bar, 1 cm.

d, Liver micro-metastases per mice were quantified by counting in ten fields.

e, H&E stained liver sections of the mice are shown, Scale bar, 100 μ m.

f, Control and KIF26B-silenced HCT116-luc cells were subcutaneously injected into BALB/C nude mice. The images of tumors in each group are presented. Scale bar, 1 cm.

g, The tumor volume was measured at indicated time, and the growth curves are shown.

h, The tumor weight was measured after the mice were sacrificed.

i, Control and KIF26B-knockdown RKO cells were subcutaneously injected into BALB/C nude mice. The images of tumors in each group are presented. Scale bar, 1 cm.

j, The tumor volume was measured at indicated time, and the growth curves are shown.

k, The tumor weight was measured after the mice were sacrificed.

Data are shown as mean \pm SD. ns, no significant, * $P < 0.05$, ** $P < 0.01$, *** $P < 0.001$, by Student's t test.

References

[1] Rokavec M, *et al.* IL-6R/STAT3/miR-34a feedback loop promotes EMT-mediated colorectal cancer invasion and metastasis. *J Clin Invest* **124**, 1853-1867 (2014).

[2] Chen RX, *et al.* N(6)-methyladenosine modification of circNSUN2 facilitates cytoplasmic export and stabilizes HMGA2 to promote colorectal liver metastasis. *Nat Commun* **10**, 4695 (2019).

[3] Wan L, *et al.* SRSF6-regulated alternative splicing that promotes tumour progression offers a therapy target for colorectal cancer. *Gut* **68**, 118-129 (2019).

[4] Grillet F, *et al.* Circulating tumour cells from patients with colorectal cancer have cancer stem cell hallmarks in ex vivo culture. *Gut* **66**, 1802-1810 (2017).

[5] Wei C, *et al.* Crosstalk between cancer cells and tumor associated macrophages is

required for mesenchymal circulating tumor cell-mediated colorectal cancer metastasis. *Mol Cancer* **18**, 64 (2019).

[6] Cespedes MV, *et al.* Orthotopic microinjection of human colon cancer cells in nude mice induces tumor foci in all clinically relevant metastatic sites. *Am J Pathol* **170**, 1077-1085 (2007).

[7] Fumagalli A, *et al.* A surgical orthotopic organoid transplantation approach in mice to visualize and study colorectal cancer progression. *Nat Protoc* **13**, 235-247 (2018).

[8] Gu S, *et al.* Knockdown of KIF26B inhibits breast cancer cell proliferation, migration, and invasion. *Onco Targets Ther* **11**, 3195-3203 (2018).

[9] Zhang H, *et al.* KIF26B, a novel oncogene, promotes proliferation and metastasis by activating the VEGF pathway in gastric cancer. *Oncogene* **36**, 5609-5619 (2017).

8. Minor, Fig. 7K, the p value and HR number are the same?

We apologize for such spelling mistakes and have corrected this error in the current manuscript (New Fig. 7k). We thank the reviewers for helping us improve our manuscript.

Fig. R22 Kaplan-Meier survival curves was analyzed and compared between patients with low (n = 129) and high (n = 129) levels of KIF26B in CRC patients from The Cancer Genome Atlas (TCGA) database. Subgroups with high- or low-KIF26B expression were sorted according to the TPM gene expression standard values. The high- or low-KIF26B are patient subgroups with KIF26B expression of top 1/3 (n = 129) or bottom 1/3 (n = 129), respectively.

REVIEWER COMMENTS

Reviewer #1 (Remarks to the Author):

The authors have addressed all my comments. However, Fig. 3d seems to be saturated and the quantification is not consistent with the dot blot shown. This need to be addressed.

Reviewer #2 (Remarks to the Author):

In assessing the revised manuscript, several major concerns remain to be addressed as listed below:

1. Functional significance of METTL3 in CRC metastasis.

It is surprising that METTL3, identified by the authors, could suppress CRC cell migration (Fig. R1a-d). Considering that almost all publications pinpoint that METTL3 not only facilitates CRC growth, but also promotes CRC metastasis through in vitro and in vivo studies (J Exp Clin Cancer Res. 2019 Sep 6; 38(1): 393. doi: 10.1186/s13046-019-1408-4. Mol Cancer. 2019 Jun 24; 18(1):112. doi: 10.1186/s12943-019-1038-7. Gastroenterology. 2020 Nov 17: S0016-5085(20)35402-0. doi: 10.1053/j.gastro.2020.11.013. Mol Cancer. 2020 Apr 3; 19(1):72. doi: 10.1186/s12943-020-01190-w. Am J Transl Res. 2020 May 15; 12(5): 1789-1806. eCollection 2020. Oncol Rep. 2020 Sep; 44(3):973-986. doi: 10.3892/or.2020.7665. Epub 2020 Jun 26. Mol Oncol. 2021 Jan 7. doi: 10.1002/1878-0261.12898), it is critical to solidify their findings. Instead of siRNA transfection, I would recommend the authors to establish CRC cells stably expressing shMETTL3 or sgMETTL3 and examine functional importance of METTL3 in CRC metastasis both in vitro and in vivo. Besides, they should repeat migration assay in cells over-expressing wild type or mutant METTL3 since mutant METTL3 expression was too high compared to wild type METTL3.

2. How F. nucleatum regulates the YAP signaling?

NF2, KIBRA, and WILLIN\FRMD6 are expressed in cytoplasm and unlikely affected by F. nucleatum directly. Any secreted proteins or metabolites produced by F. nucleatum participated? If so, mutant F. nucleatum could be constructed for further investigation.

3. Validation of m6A-seq data using m6A-RT-qPCR.

Not clear how the authors define significant m6A peaks, although they mentioned in the method section that significant peaks were selected by MACS2 (Threshold value = 1). How about q-value (minimum FDR) as cutoff? Besides, they should validate their findings by m6A-RT-qPCR. Whether m6A enrichment of KIF26B mRNA was altered after F. nucleatum infection or modulation of METTL3 (depletion, wild type and mutant METTL3).

4. Fig. 7A-D, wouldn't be mouse models of CRC (APC min or AOM) more suitable? How about METTL3 mRNA expression?

5. Fig. 7F-L, it was not clear why the authors included only 258 CRC patients from TCGA cohort for analysis.

Reviewer #3 (Remarks to the Author):

Most concerns have been addressed. I would suggest the authors include the invasion data, at least as supplemental data.

REVIEWER COMMENTS

Reviewer #1 (Remarks to the Author):

The authors have addressed all my comments. However, Fig. 3d seems to be saturated and the quantification is not consistent with the dot blot shown. This need to be addressed.

We really thank the reviewers for helping us improve our manuscript. We apologize for such typo mistake and have corrected this error in the current manuscript (**Fig. 3d**).

Reviewer #2 (Remarks to the Author):

In assessing the revised manuscript, several major concerns remain to be addressed as listed below:

1. Functional significance of METTL3 in CRC metastasis.

It is surprising that METTL3, identified by the authors, could suppress CRC cell migration (Fig. R1a-d). Considering that almost all publications pinpoint that METTL3 not only facilitates CRC growth, but also promotes CRC metastasis through in vitro and in vivo studies (J Exp Clin Cancer Res. 2019 Sep 6;38(1):393. doi: 10.1186/s13046-019-1408-4. Mol Cancer. 2019 Jun 24;18(1):112. doi: 10.1186/s12943-019-1038-7. Gastroenterology. 2020 Nov 17:S0016-5085(20)35402-0. doi: 10.1053/j.gastro.2020.11.013. Mol Cancer. 2020 Apr 3;19(1):72. doi: 10.1186/s12943-020-01190-w. Am J Transl Res. 2020 May 15;12(5):1789-1806. eCollection 2020. Oncol Rep. 2020 Sep;44(3):973-986. doi: 10.3892/or.2020.7665. Epub 2020 Jun 26. Mol Oncol. 2021 Jan 7. doi: 10.1002/1878-0261.12898), it is critical to solidify their findings. Instead of siRNA transfection, I would recommend the authors to establish CRC cells stably expressing shMETTL3 or sgMETTL3 and examine functional importance of METTL3 in CRC metastasis both in vitro and in vivo.

We thank the reviewer for raising this concern. As the reviewer mentioned, there have been many studies explored the functions of METTL3 in CRC. METTL3 not only affected the proliferation of CRC cells, but also altered cell migration ability. However, the function of METTL3 in CRC metastasis is actually paradoxical [1, 2, 6, 7]. According to the reviewer's suggestions, we established HCT116 cells stably

expressing sgRNA targeting METTL3 (**Fig. R1a**) or shRNA targeting METTL3 (**Fig. R1d**). Interestingly, knockout or knockdown of METTL3 significantly inhibited the proliferation of HCT116 cells (**Fig. R1b, e**), which is consistent with previous reports [1-5]. The indicated cells were further subjected to the transwell migration assay. The data showed the depletion of METTL3 indeed promoted the migration ability of HCT116 cells (**Fig. R1c, f**), which is consistent with our previous data of siRNA transfection.

In this study, we uncover that under the stimulation of *F. nucleatum*, the METTL3 expression in CRC cells, as well as the m⁶A levels decreased through the activation of YAP signaling, contributing to *F. nucleatum*-induced CRC aggressiveness. Considering the functional complexity of METTL3 in tumor progression, although we indeed have observed that knockdown of METTL3 can facilitate CRC cell migration upon *F. nucleatum* existence, we don't want to get conclusions about what it functions in the absence of *F. nucleatum* and will not claim this point in our manuscript. Definitely, the complicated regulational mechanisms and functions of METTL3, as well as m⁶A modifications in cancer, deserve further studies in future.

Besides, they should repeat migration assay in cells over-expressing wild type or mutant METTL3 since mutant METTL3 expression was too high compared to wild type METTL3.

We thank the reviewer for helping us improve our manuscript. According to the reviewer's suggestion, we repeat the assay. The indicated cells expressed ectopic mutant METTL3 comparable to the wild type METTL3 (**Fig. R1g**). We repeated the migration assay. Ectopic expression of wild type METTL3, but not its catalytic mutant, obviously reversed the promotion of HCT116 cells migration stimulated by *F. nucleatum* treatment (**Fig. R1h**). We have added the Fig. R2g and R2h as the New Supplementary Fig. 1h and 1j in the current manuscript.

Fig. R1: The function of METTL3 with or without *F. nucleatum* infection in CRC cells. **a, d**, Western blot was performed to detect the expression of METTL3 in HCT116 cells infected with sgMETTL3 (**a**) or shMETTL3 (**d**). **b, e**, HCT116 cells with indicated treatments were subjected to colony formation analysis (Left). The colony were quantified by counting in three fields (Right). Scale bar, 1 cm. **c, f**, HCT116 cells with indicated treatments were subjected to transwell migration analysis (Left). Migrated cells were quantified by counting in five fields (Right). Scale bar, 100 μ m. **g**, The HCT116 cells were transfected with wild-type METTL3, or catalytic mutant (aa395-398, DPPW-APPA) METTL3, or control pcDNA3.1 plasmids and treated with *F. nucleatum* or PBS. Western blot analysis of METTL3 levels were performed. **h**, he HCT116 cells with indicated treatments were applied for transwell migration analysis. Representative images of migrated cells were shown (Left). The migrated cells were quantified in five fields. Scale bar, 100 μ m. Data were quantified using ImageJ software

and normalized to the loading control. Data are shown as mean \pm SD. *** $P < 0.001$, **** $P < 0.0001$, by Student's t test.

References

- [1] Li T, *et al.* METTL3 facilitates tumor progression via an m(6)A-IGF2BP2-dependent mechanism in colorectal carcinoma. *Mol Cancer* 18, 112 (2019).
- [2] Zhou D, *et al.* METTL3/YTHDF2 m6A axis accelerates colorectal carcinogenesis through epigenetically suppressing YPEL5. *Mol Oncol* 15, 2172-2184 (2021).
- [3] Chen H, *et al.* RNA m6A methyltransferase METTL3 facilitates colorectal cancer by activating m6A-GLUT1-mTORC1 axis and is a therapeutic target. *Gastroenterology*, (2020).
- [4] Shen C, *et al.* m(6)A-dependent glycolysis enhances colorectal cancer progression. *Mol Cancer* 19, 72 (2020).
- [5] Song X, *et al.* N6-methyladenosine methyltransferase METTL3 promotes colorectal cancer cell proliferation through enhancing MYC expression. *Am J Transl Res* 12, 5 (2020)
- [6] Peng W, *et al.* Upregulated METTL3 promotes metastasis of colorectal Cancer via miR-1246/SPRED2/MAPK signaling pathway. *J Exp Clin Cancer Res* 38, 393 (2019).
- [7] Deng R, *et al.* m(6)A methyltransferase METTL3 suppresses colorectal cancer proliferation and migration through p38/ERK pathways. *Oncotargets Ther* 12, 4391-4402 (2019).

2. How *F. nucleatum* regulates the YAP signaling?

NF2, KIBRA, and WILLIN\FRMD6 are expressed in cytoplasm and unlikely affected by *F. nucleatum* directly. Any secreted proteins or metabolites produced by *F. nucleatum* participated? If so, mutant *F. nucleatum* could be constructed for further investigation.

We thank the reviewer for raising this interesting topic. In this study, we revealed that *F. nucleatum* can reduce the METTL3 expression through activation of YAP signaling in the CRC cells. Mechanistically, our results showed that *F. nucleatum* treatment obviously inhibited the phosphorylation of MST, as well as its upstream factors NF2, KIBRA, and WILLIN\FRMD6, suggesting that *F. nucleatum* may inactivate the HIPPO signaling pathway. Regarding NF2, KIBRA, and WILLIN\FRMD6 are

localized in cytoplasm, *F. nucleatum* was reported to adhere and invade into epithelial cells [1, 2, 3]. A recent study confirmed the invasive potential of *F. nucleatum* into HCT116 cells by fluorescence microscopy and flow cytometry [4]. On the other hand, *F. nucleatum* may secrete virulence factors or metabolites [5, 6] to affect host cells. However, the exact upstream mechanisms of HIPPO pathway, such as how the extracellular events stimulate NF2, KIBRA, WILLIN\FRMD6, and MST complex, is still a mystery in the HIPPO-YAP field. We fully agree that elucidation of the mechanism for how the gut microbes affect the host HIPPO-YAP pathway will be great interesting and will have a more comprehensive understanding of the upstream mechanism of the HIPPO-YAP pathway. However, exploring the upstream mechanism for HIPPO pathway will be a challenging huge work which deserves another subtle story in future. We have added this part of the content in the Discussion section of the current manuscript.

References

- [1] Abed J, *et al.* Fap2 Mediates Fusobacterium nucleatum Colorectal Adenocarcinoma Enrichment by Binding to Tumor-Expressed Gal-GalNAc. *Cell Host Microbe* 20, 215-225 (2016).
- [2] Rubinstein MR, *et al.* Fusobacterium nucleatum promotes colorectal carcinogenesis by modulating E-cadherin/beta-catenin signaling via its FadA adhesin. *Cell Host Microbe* 14, 195-206 (2013).
- [3] Nakagaki H, *et al.* Fusobacterium nucleatum envelope protein FomA is immunogenic and binds to the salivary statherin-derived peptide. *Infect Immun* 78, 1185-92 (2010).
- [4] Casasanta MA, *et al.* Fusobacterium nucleatum host-cell binding and invasion induces IL-8 and CXCL1 secretion that drives colorectal cancer cell migration. *Sci Signal* 13 (2020).
- [5] Kostic AD, *et al.* Fusobacterium nucleatum potentiates intestinal tumorigenesis and modulates the tumor-immune microenvironment. *Cell Host Microbe* 14, 207-215 (2013).
- [6] Kapatral V, *et al.* Genome sequence and analysis of the oral bacterium Fusobacterium nucleatum strain ATCC 25586. *J Bacteriol* 184, 2005-2018 (2002).

3. Validation of m6A-seq data using m6A-RT-qPCR.

Not clear how the authors define significant m6A peaks, although they mentioned in the method section that significant peaks were selected by MACS2 (Threshold value = 1). How about q-value (minimum FDR) as cutoff? Besides, they should validate their findings by m6A-RT-qPCR. Whether m6A enrichment of KIF26B mRNA was altered after *F. nucleatum* infection or modulation of METTL3 (depletion, wild type and mutant METTL3).

We apologize for making you confused. Significant m⁶A peaks with FDR < 0.05 were screened for subsequent analysis. We have added this information in the revised METHODS section.

We appreciate the reviewer's instructive suggestions. According to the reviewer's suggestion, we designed primers for m⁶A-RT-qPCR in the region containing the changed m⁶A peak, which were annotated in our schema diagram (Fig. R2a). m⁶A-RT-qPCR assay was performed and the result showed that the m⁶A enrichment of *KIF26B* mRNA was indeed reduced upon *F. nucleatum* treatment (Fig. R2b). We have added the Fig R2 as the New Fig. 5c and 5d in the current manuscript.

Fig. R2: *F. nucleatum* reduced the m⁶A levels of *KIF26B* mRNA. a, m⁶A RIP-seq data showed a significant decrease of m⁶A peaks around the stop codon of *KIF26B* mRNA upon *F. nucleatum* treatment. Squares marked decline of m⁶A peaks in HCT116 cells co-cultured with *F. nucleatum*. The region for m⁶A-RT-qPCR detection was annotated. b, m⁶A RIP-qPCR analysis of *KIF26B* mRNA in the control and *F. nucleatum* treatment HCT116 cells. Data are shown as mean ± SD. *** *P* < 0.001, by Student's *t* test.

4. Fig. 7A-D, wouldn't be mouse models of CRC (APC min or AOM) more suitable? How about METTL3 mRNA expression?

We appreciate the reviewer's good suggestion. Actually, we also have the same idea with the reviewer. Since the m⁶A modification, as well as METTL3, may be altered

as response to stress stimulation, we were not sure whether AOM treatment will interfere the results, thus we applied the wild-type C57BL/6 mice to confirm the function of *F. nucleatum* under physiological condition. Follow the reviewer's instructive suggestions, here, we use APC^{Min/+} model to verify our findings in this study. Similar to the previous model, the APC^{Min/+} mice were pretreated with 2 mg/ml streptomycin by gavage administration for 3 days. Then, the mice were administrated with 10⁹ CFU *F. nucleatum* or *E. coli* every day. After 15 days, mice were sacrificed. The following qRT-PCR analysis showed that upon exposure to *F. nucleatum* (**Fig. R3a**), the *METTL3* mRNA expression in colorectum tissues significantly reduced compared to that from the *E. coli*-treated mice (**Fig. R3b**), while the mRNA levels of *KIF26B* were significantly increased (**Fig. R3c**). We have added the **Fig. R3a-c** as the **New Supplementary Fig. 8c-e** in the current manuscript.

For *F. nucleatum*-treatment C57BL/6 mice model, consistent with protein levels, the mRNA levels of *METTL3* were significantly downregulated in *F. nucleatum*-treated group (**Fig. R3d**). We have added the **Fig. R3d** as the **New Fig. 7c** in the current manuscript.

Fig. R3: *F. nucleatum* is correlated with *METTL3* and *KIF26B* expressions in animal models. **a**, Quantitative RT-PCR analysis of *F. nucleatum* in stool from the indicated APC^{Min/+} mice (n = 3). Data are presented as log₂ value of *F. nucleatum* 16S normalized to universal *Eubacteria* 16S. **b**, **c**, Quantitative RT-PCR analysis of *METTL3* (**b**) and *KIF26B* (**c**) mRNA expression in colorectum tissues from the indicated APC^{Min/+} mice (n = 3). The data after presented after normalization to the GAPDH control. **d**, Quantitative RT-PCR analysis of *F. nucleatum* in stool from the indicated C57BL/6 mice (n = 6). Data are presented as log₂ value of *F. nucleatum* 16S normalized to universal *Eubacteria* 16S. Data are shown as mean ± SD. * P < 0.05, **

$P < 0.01$, by Student's t test.

5. Fig. 7F-L, it was not clear why the authors included only 258 CRC patients from TCGA cohort for analysis.

We apologize for not making this clear in the manuscript and thank the reviewer for raising this point. In the TCGA database, the colorectal cancer transcriptome sequencing dataset contains 380 colorectal cancer patients with TPM gene expression standard values and complete pathological information. We ranked these patients according to the expression levels of KIF26B. We defined the top 1/3 patients as the KIF26B high expression group ($n = 129$), and the bottom 1/3 patients as the KIF26B low expression group ($n = 129$). These two groups of patients were applied for Kaplan-Meier survival curves analysis and multivariable analysis. We have made this clear in the revised manuscript and also added these details in the revised METHODS section and figure Legend.

Reviewer #3 (Remarks to the Author):

Most concerns have been addressed. I would suggest the authors include the invasion data, at least as supplemental data.

We really appreciate the reviewer's advices. We have added these invasion data in our current manuscript (**New Supplementary Fig. 1b, d and New Supplementary Fig. 6d, g**).

REVIEWERS' COMMENTS

Reviewer #2 (Remarks to the Author):

Comments have been well addressed.